# Plant Trans-Golgi Network/Early Endosome pH regulation requires Cation Chloride Cotransporter (CCC1)

**Daniel W McKay[1], Heather E McFarlane[2,3], Yue Qu[1], Apriadi Situmorang[1], Matthew Gilliham[1], Stefanie Wege[1]***

[1]School of Agriculture, Food and Wine, Waite Research Institute, ARC Centre of Excellence in Plant Energy Biology, University of Adelaide, Waite Campus, Adelaide, Australia; [2]School of Biosciences, University of Melbourne, Melbourne, Australia; [3]Department of Cell and Systems Biology, University of Toronto, Toronto, Canada

**Abstract** Plant cells maintain a low luminal pH in the trans-Golgi-network/early endosome (TGN/EE), the organelle in which the secretory and endocytic pathways intersect. Impaired TGN/EE pH regulation translates into severe plant growth defects. The identity of the proton pump and proton/ion antiporters that regulate TGN/EE pH have been determined, but an essential component required to complete the TGN/EE membrane transport circuit remains unidentified – a pathway for cation and anion efflux. Here, we have used complementation, genetically encoded fluorescent sensors, and pharmacological treatments to demonstrate that *Arabidopsis* cation chloride cotransporter (CCC1) is this missing component necessary for regulating TGN/EE pH and function. Loss of CCC1 function leads to alterations in TGN/EE-mediated processes including endocytic trafficking, exocytosis, and response to abiotic stress, consistent with the multitude of phenotypic defects observed in *ccc1* knockout plants. This discovery places CCC1 as a central component of plant cellular function.

*For correspondence:
stefanie.wege@adelaide.edu.au

## Editor's evaluation

This paper reports compelling evidence that the single isoform of cation chloride cotransporter (CCC1) encoded in *Arabidopsis thaliana* provides a cation and anion efflux mechanism to regulate pH in the trans-Golgi/Early Endosome Network.

## Introduction

The transport of ions across biological membranes is fundamental to diverse cellular functions such as energy production, nutrient uptake and distribution, growth, signalling, and osmoregulation. Ion influx and efflux must be coordinated to regulate osmotic potential, and to maintain charge balance within and across cellular compartments. Ion efflux out of cells contributes towards processes as varied as neuronal function in animals, heavy metal tolerance in bacteria, and salinity tolerance in plants. Ion efflux out of subcellular compartments is also critical for core cellular functions such as chloroplastic photosynthetic efficiency in plants (*Correa Galvis et al., 2020*).

Cation chloride cotransporters (CCCs) are cation and anion symporters that are present in all domains of life, whether archaea, animal, yeast, or plant. Chloride (Cl⁻) is transported by CCCs electroneutrally across membranes with the cations of sodium (Na⁺) and/or potassium (K⁺). A single CCC exists in the archaeon, *Methanosarcina acetivorans*, which does not share a high degree of similarity to eukaryotic CCCs (*Hartmann and Nothwang, 2015*). In animals, CCCs have been separated into

two subfamilies: the Cl⁻ and K⁺ efflux symporters (KCCs), and a second subfamily with two clades of CCC, the Cl⁻ and Na⁺/K⁺ influx symporters (NKCCs), and the Cl⁻ and Na⁺ influx symporters (NCCs). Vertebrates typically harbour several CCC isoforms from each subfamily. For example, humans (*Homo sapiens*) have seven CCC isoforms (4 KCC, 3 NKCC/NCCs) and zebrafish (*Danio rerio*) have 12 CCCs in total (*Hartmann et al., 2014*). In non-vascular plants there are two clades of CCC, CCC1 and CCC2, with some bryophytes, such as the moss *Physcomitrium patens* (formerly *Physcomitrella patens*) with up to seven isoforms (*Henderson et al., 2018*). In contrast, in flowering plants (angiosperms), CCCs are represented by only the CCC1 clade, with very few isoforms per plant. For instance, rice (*Oryza sativa*), contains two CCC1s, and *Arabidopsis* (*Arabidopsis thaliana*) has only one CCC1.

In vertebrates, CCCs are involved in diverse but specific cellular processes including osmoregulation in kidneys and neurons (regulation of cell volume), potassium secretion in muscles and regulation of Cl⁻ concentrations in neural tissue (*Blaesse et al., 2009*; *Hartmann and Nothwang, 2015*). NKCCs and NCCs drive influx of ions across the cell membrane, often by utilising large extracellular Na⁺ concentrations to drive transport of Cl⁻ and K⁺, or just Cl⁻ in the case of NCC, whereas KCCs utilise K⁺ gradients to mediate Cl⁻ efflux out of the cytosol (*Blaesse et al., 2009*; *Arroyo et al., 2013*). The importance of Cl⁻ management and osmoregulation for cell and organ function is highlighted by the large number of human diseases and conditions associated with CCC malfunction including schizophrenia, deafness, muscle spasticity, epilepsy, Andermans syndrome (neuropathy), and, Bartter and Gitelman syndrome (both characterised by kidney disfunction) (*Simon et al., 1996*; *Kunchaparty et al., 1999*; *Howard et al., 2002*; *Boettger et al., 2002*; *Rivera et al., 2002*; *Boulenguez et al., 2010*; *Kim et al., 2012*).

The single CCC1 protein in the model plant *Arabidopsis* also appears to have a crucial, but unexplained, role in plant function. When *Arabidopsis* CCC1 is genetically ablated, *ccc1* plants exhibit complex and severe phenotypic defects which include a large reduction in overall plant size, a bushy appearance characterised by increased axillary shoot outgrowth, frequent stem necrosis, very low fertility, alterations in pathogen response, changes in cell wall composition, and changes in seed ion concentrations (*Johnson et al., 2004*; *Colmenero-Flores et al., 2007*; *McDowell et al., 2013*; *Henderson et al., 2018*; *Han et al., 2020*). *Arabidopsis ccc1* also displays changes in ion distribution. Hydroponically grown *ccc1* plants accumulate more Cl⁻ in shoots under both control and 50 mM NaCl conditions while soil-grown *ccc1* plants accumulate more Cl⁻ in shoots when exposed to 50 mM Cl⁻ salts (*Colmenero-Flores et al., 2007*; *Henderson et al., 2015*). Similarly, roots of rice *Osccc1.1* plants also had altered ion accumulation profiles under both control and saline conditions (*Chen et al., 2016*). While rice harbours two CCC1 proteins, *Arabidopsis* is the ideal system for investigating the role of CCCs in higher plants due to the presence of only a single CCC homologue, eliminating the possibility for functional redundancy (*Henderson et al., 2018*).

Plant, fungi, and animal CCC subcellular localisations are different. While many of the animals' CCCs have been found on the cell membrane (plasma membrane [PM]), the yeast (*Saccharomyces cerevisiae*) CCC homologue, VHC1, is localised not on the exterior membrane of the cell, but on the yeast vacuolar membrane (*Petrezselyova et al., 2013*). Independent proteomic studies in *Arabidopsis* have revealed that the *Arabidopsis* CCC1 is highly abundant in the trans-Golgi network/early endosome (TGN/EE) (*Sadowski et al., 2008*; *Drakakaki et al., 2012*; *Groen et al., 2014*; *Nikolovski et al., 2014*), with the latest study defining CCC1 as a TGN/EE marker protein (*Parsons et al., 2019*). Heterologous expression studies in tobacco (*Nicotiana benthamiana*) epidermal cells also suggested an endomembrane localisation of the *Arabidopsis* CCC1 using a translational fluorescent protein fusion (*Henderson et al., 2015*). This raises the question of the role of CCC1 in this organelle. Several transport proteins in the TGN/EE have been identified, with mutants of the transport proteins involved in TGN/EE pH regulation exhibiting similar phenotypic aspects to *ccc1* knockouts. CCC1 is therefore a candidate ion transporter that might contribute to TGN/EE pH regulation.

The plant TGN/EE is an organelle with a complex cellular role. One of its key roles is sorting and delivering proteins to the apoplast, PM, and vacuole (*Dettmer et al., 2006*; *Viotti et al., 2010*; *Sze and Chanroj, 2018*). Four TGN/EE Cl⁻ and K⁺ transporters have previously been identified: the H⁺/K⁺ antiporters NHX5 and NHX6, and the H⁺/Cl⁻ antiporters CLCd and CLCf. These antiporters import K⁺ and Cl⁻ into the TGN/EE in exchange for luminal protons and work in conjunction with the TGN/EE V-ATPase complex to regulate luminal pH (*Bassil et al., 2011*; *Reguera et al., 2015*; *Scholl et al., 2021*). Finely tuned luminal pH is required for the cellular function of the TGN/EE (*Martinière et al.,*

*2013*; *Luo et al., 2015*; *Reguera et al., 2015*). Plants with reduced V-H⁺-ATPase activity (e.g. *det3* mutants) have an elevated TGN/EE pH and exhibit severe developmental growth defects, which is accompanied by alterations in exocytosis of PM proteins, such as BRI1, defects in cell division and expansion and hypersensitivity to salt (*Schumacher et al., 1999*; *Dettmer et al., 2006*; *Brüx et al., 2008*; *Krebs et al., 2010*; *Luo et al., 2015*). Double *nhx5/nhx6* mutants have similar, but not identical, growth defects to *det3*, with reduced cell length, a large decrease in overall plant size, and increased salt sensitivity of root growth and seed germination (*Bassil et al., 2011*; *Reguera et al., 2015*). As in *det3*, trafficking of BRI1 is altered in *nhx5/nhx6*; however, only recycling is affected while secretion to the PM is not (*Dragwidge et al., 2019*). In addition, the PM abundance of the membrane integrated proteins, PIN1 and PIN2, are reduced in *nhx5/nhx6* (*Dragwidge et al., 2018*). However, in contrast to *det3*, the *nhx5/nhx6* TGN/EE lumen is hyperacidic, consistent with the proposed role of NHX in exporting protons out of the TGN/EE (*Luo et al., 2015*; *Reguera et al., 2015*). The hyperacidity of *nhx5/nhx6* TGN/EE lumen results in mis-sorting of vacuolar proteins due to altered binding of targets by the vacuolar sorting receptor VSR1;1 (*Reguera et al., 2015*). A *clcd* single knockout has a mild increase in pathogen sensitivity. The less severe phenotype of this knockout might be a result of the two TGN/EE resident CLC proteins, which are likely to exhibit functional redundancy similar to the two TGN/EE NHXs. Collectively, these observations demonstrate that a typical consequence of alterations in TGN/EE lumen pH regulation are defects in protein trafficking, particularly in protein exocytosis, changes in salt tolerance, and defects in cell elongation and division, resulting in severe impacts on plant growth.

The current model of TGN/EE pH regulation, as outlined above, is incomplete; it currently includes a proton pump (V-H⁺-ATPase) and anion- and cation-proton exchangers (CLC, NHX) (*Sze and Chanroj, 2018*), but a transport protein that mediates either cation or anion efflux has not yet been identified. Both import and export of ions are crucial components of dynamic cellular and organellar ion regulation. If both do not occur, cation and anion import into the TGN/EE would cease once the gradient becomes too high, which would inhibit the function of the antiporters. Plant CCC1s, with their cation and anion symport function and their residency in the TGN/EE, are ideal candidates to fill this missing role and complete the circuit through export of Cl⁻ and K⁺ from the TGN/EE.

Here, we characterised the role of *Arabidopsis* CCC1 in the TGN/EE. We demonstrate that TGN/EE-localised CCC1 rescues cellular defects of *ccc1* knockouts in *Arabidopsis*, and that CCC1 impacts TGN/EE luminal pH and osmotic regulation. Genetic ablation of CCC1 function leads to defects in TGN/EE-dependent processes, specifically, *ccc1* plants exhibit reduced rates of exocytosis and endocytic trafficking, typical of altered TGN/EE pH regulation. As such, we propose that CCC1 is a missing core component of the ion- and pH-regulating machinery of the TGN/EE.

## Results

### CCC1 *is ubiquitously expressed*

Previous reports on *CCC1* expression are contradictory. Promoter-GUS studies indicated that *CCC1* expression was restricted to specific tissues, such as root stele or hydathodes and pollen, while RNA transcriptomic studies, including single-cell RNAseq, suggest expression occurs in a broader range of cell types (*Colmenero-Flores et al., 2007*; *Wendrich et al., 2020*). To clarify the tissue expression pattern of *CCC1*, we transformed wildtype *Arabidopsis* plants with a 2 kb genomic DNA sequence upstream of the *CCC1* coding region driving the expression of nuclear-localised triple Venus (a bright variant of the yellow fluorescent protein) or β-glucuronidase (GUS), named *CCC1prom::Venus* and *CCC1prom::GUS*, respectively. Combined analysis of fluorescence and GUS staining revealed that *CCC1* is expressed in all cell types, including all root cells, hypocotyl, leaf and stem epidermis, guard cells and trichomes, as well as mesophyll cells and all flower parts (*Figure 1*). *CCC1* promoter activity reported by Venus fluorescence, or by GUS-activity, was slightly different despite use of the identical promoter sequence. For instance, fluorescence was detectable in root cortex and epidermis cells, including root hairs, and in the gynoecium, while GUS staining did not indicate expression in these cells. This is likely due to the greater sensitivity of the Venus fluorescence method.

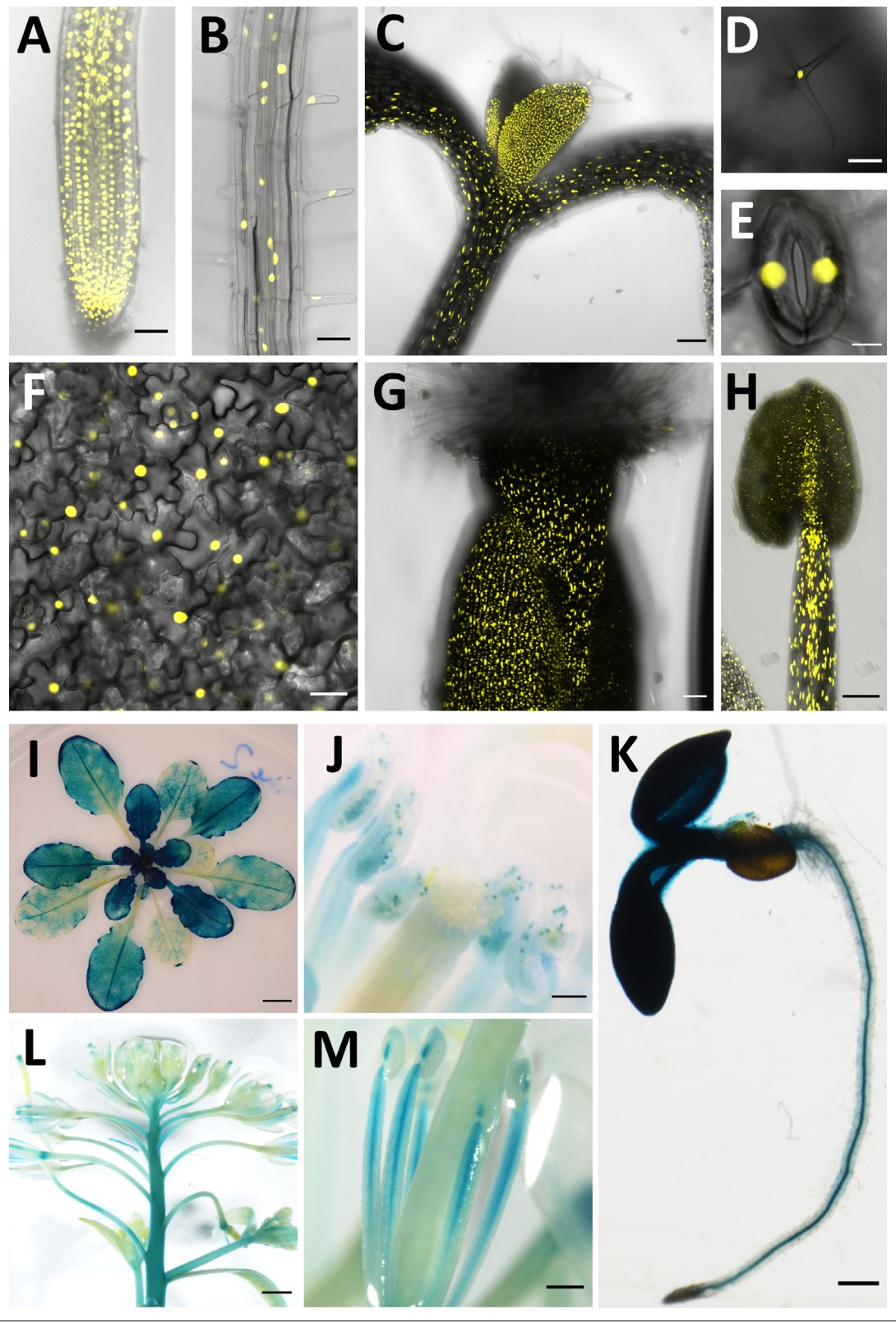

**Figure 1.** *CCC1* is expressed ubiquitously. *CCC1* promoter-driven expression of either NLS-Venus (yellow, bright YFP variant with a nuclear localisation signal) or ß-glucuronidase (blue GUS staining). (**A–H**) NLS-Venus expression indicating CCC1 promoter activity in all root cells, including (**A**) the root tip, (**B**) root epidermal cells; (**C**) hypocotyl; all leaf cells including (**D**) trichomes, (**E**) guard cells, and (**F**) leaf epidermal and mesophyll cells; and reproductive organs (**G**) gynoecium, and (**H**) stamen tissues. (**I–M**) GUS staining indicating promoter activity predominantly in (**I**) younger leaves, (**J**) pollen, (**K**) root

*Figure 1 continued on next page*

*Figure 1 continued*

stele, (**L**) floral stem, and (**M**) stamen. Scale bars are 50 µm (images **A**, **B, G**), 100 µm (images **C**, **D, H, J**), 5 µm (image **E**), 20 µm (image **F**), 5 mm (image **I**), 200 µm (images **K**, **M**) and 1000 µm (image **L**).

## Loss of CCC1 results in growth defects in root cells

*ccc1* knockout plants are severely affected in their growth, including a reduced shoot size and shorter primary roots (*Figure 2A*, and previously quantified by *Colmenero-Flores et al., 2007*; *Henderson et al., 2015*). We investigated the origin of the root phenotype of *ccc1* at a cellular level and found that CCC1 function is required for cell elongation, contributing to the reduced root length in knockout mutants. *ccc1* develop both shorter root epidermal cells, and shorter root hairs (*Figure 2B–E*) and a complete lack of collet hairs (*Figure 2F*). Collet hairs are epidermal root hairs formed in some plant species in the transition zone between the root and the hypocotyl (*Sliwinska et al., 2015*). In addition, *ccc1* root hairs displayed branching and bulging (*Figure 2—figure supplement 1*), although at an extremely low frequency; while ruptured root hairs were never observed.

Independent of the defect of root hair elongation, *ccc1* plants frequently developed root hairs in cell files that usually exclusively contain atrichoblasts (non-root hair cells) in the wildtype. These

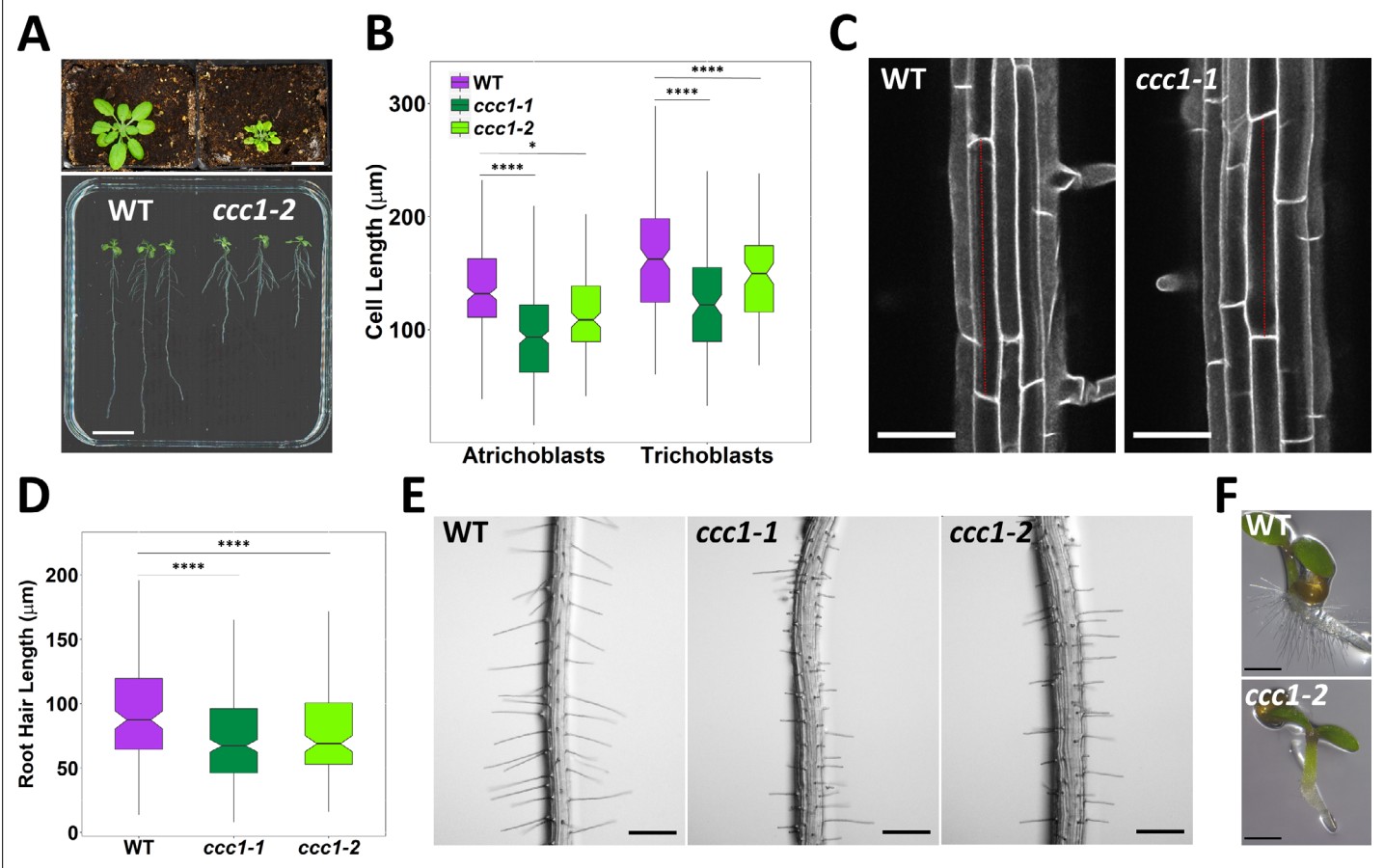

**Figure 2.** *ccc1* plants show defects in cell elongation. (**A**) Top image, *ccc1* (right) have smaller shoots and deformed leaves compared to wildtype (left) plants. Plants grown 26 days in short day, scale bar is 2 cm. Bottom image, *ccc1* (right) have shorter primary roots compared to wildtype (left) plants. Plants grown 14 days in long day, scale bar is 2 cm. (**B–C**) Root epidermal cells are shorter in *ccc1*. n > 13 plants. Images are maximum intensity projections of cell wall autofluorescence. Scale bars are 50 µm. (**D–E**) *ccc1* plants have shorter root hairs. n > 900 root hairs of >30 plants. Scale bars are 200 µm. (**F**) *ccc1* plants do not develop collet hairs. Scale bars are 500 µm. Boxplots show range; median, first and third quartile are indicated. One-way ANOVA used to determine p. * indicates p < 0.05, **** indicates p < 0.0001.

The online version of this article includes the following source data and figure supplement(s) for figure 2:

**Source data 1.** Data analysed and presented in *Figure 2* and supplement.

**Figure supplement 1.** CCC1 is important for collet hair elongation and normal trichoblast development.

ectopic root hairs are formed in cells that have trichoblast identity, as they express the trichoblast marker *PRP3::H2B-2× mCherry* (*Marquès-Bueno et al., 2016*). Fluorescence of this nuclear-localised marker occurred in multiple adjacent cells in *ccc1*, which did not occur in wildtype plants (*Figure 2— figure supplement 1*).

### Functional GFP-CCC1 localises to the endomembrane system

We had previously localised *Arabidopsis* CCC1-GFP to the Golgi and TGN/EE in transient expression assays in *N. benthamiana* (*Henderson et al., 2015*). In addition, several proteomic studies corroborate that CCC1 is present in this organelle (*Sadowski et al., 2008*; *Drakakaki et al., 2012*; *Groen et al., 2014*; *Nikolovski et al., 2014*), with one study identifying CCC1 as a high-confidence TGN/EE resident protein (*Parsons et al., 2019*). Therefore, we sought to investigate the role of CCC1 in this organelle. We generated an N-terminally tagged GFP-CCC1 construct, which, when stably expressed in *ccc1* plants, was able to complement the root hair phenotype. To express this construct, we used the *EXP7* (*Expansin7*) trichoblast-specific promoter, which drives expression within root epidermal cells. This approach was adopted after many attempts to generate plants that express the CCC1 protein with different tags and different fusion orientations using the native promoter (*Supplementary file 1a*). Approaches included the use of different protein linkers between CCC1 and the different tags, green and red fluorescent proteins, the smaller FLAG tag, and multiple tag locations, including

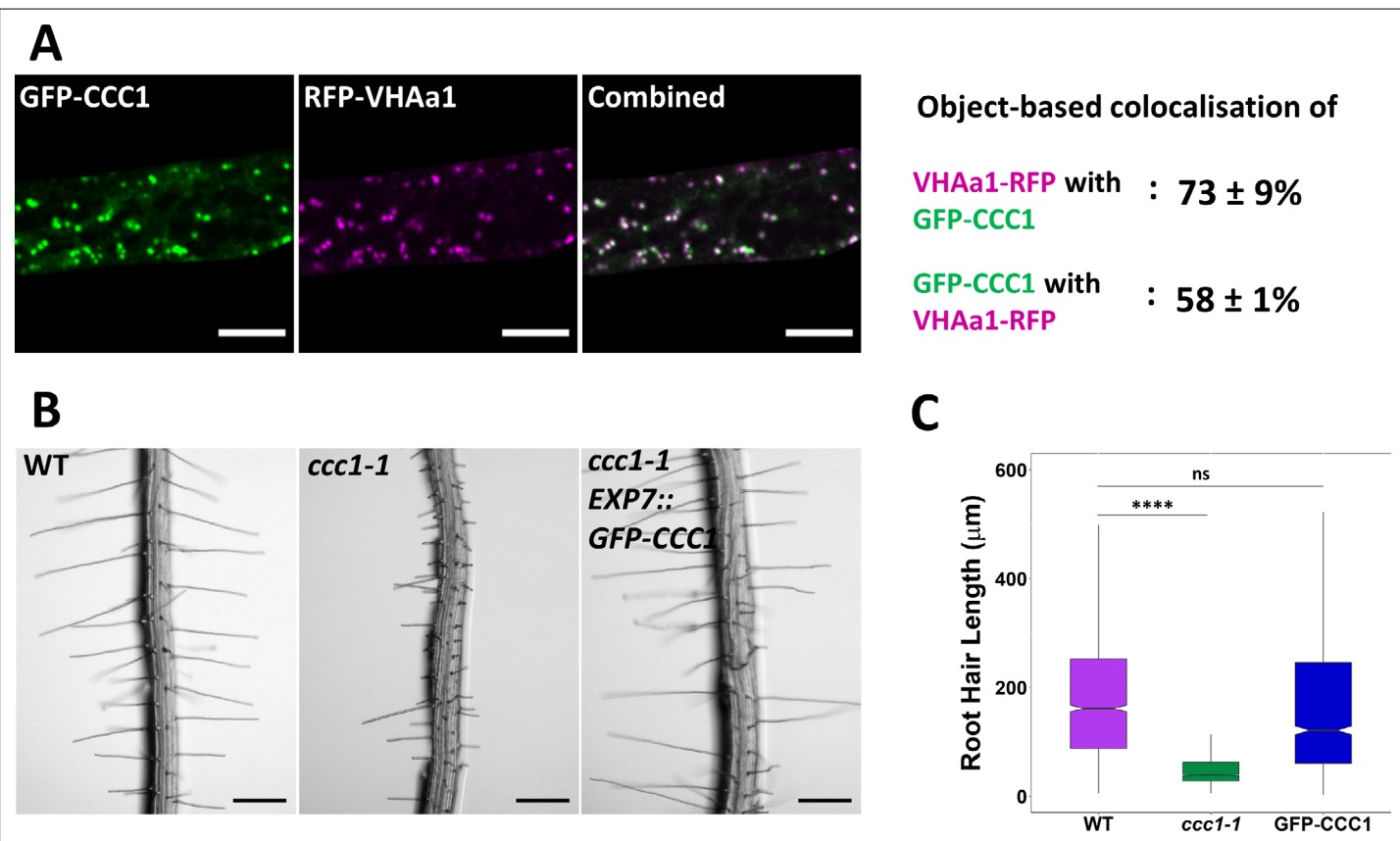

**Figure 3.** Stably expressed GFP-CCC1 is functional and localised to the trans-Golgi-network/early endosome (TGN/EE). (**A**) GFP-CCC1 (green) and VHAa1-RFP (magenta) colocalise. Colocalisation was calculated using DiAna object-based colocalisation plugin in ImageJ; Pearson's coefficient was also calculated as 0.86 ± 0.055. Error is standard deviation. n = 15 cells of five plants. Scale bars are 10 µm. Images are single representative optical sections from a stack. (**B–C**) Expression of GFP-CCC1 rescues *ccc1* root hair length defects. n > 1300 root hairs. Scale bars are 200 µm. Boxplot shows range; median, first and third quartile are indicated. One-way ANOVA used to determine p. **** indicates p < 0.0001.

The online version of this article includes the following source data and figure supplement(s) for figure 3:

**Source data 1.** Data analysed and presented in *Figure 3* and supplement.

**Figure supplement 1.** GFP-CCC1 is localised to the endomembrane system and neither pharmacological treatment nor osmotic shock alters localisation.

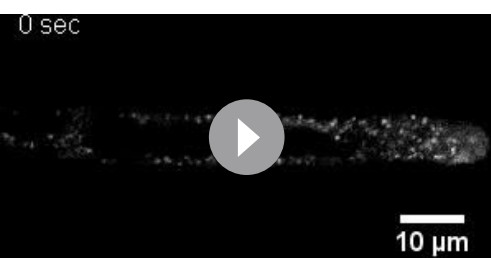

**Video 1.** Time-lapse showing movement of GFP-CCC1 labelled subcellular compartments in a root hair. GFP-CCC1 localises to motile intracellular organelles, expression driven with *EXP7* promoter. Time series of the root hair was taken through the centre plane of the root hair. 20 s shown per second at 10 frames per second.

https://elifesciences.org/articles/70701/figures#video1

both termini and internal placements. No transformants were recovered with a terminally tagged protein expressed using the native promoter. This difficulty in obtaining transformed plants might suggest that terminal tagging interferes with CCC1 function in embryonic or meristematic tissue where CCC1 is active. Generation of an antibody was also not successful. Internal tagging might have disrupted protein folding, despite the selection of tagging sites according to a protein model made using Swissport since transformants were recovered, but no fluorescence could be detected and the *ccc1* knockout phenotype was not complemented in those plants. We therefore decided to express GFP-CCC1 in a mature cell type, in which we had identified a clear phenotypic defect in *ccc1* – trichoblasts. *GFP-CCC1* expression in these epidermal cells was successful and complemented the short root hair phenotype of *ccc1* knockout plants (*Figure 3*). GFP-CCC1 expression driven by the *EXP7* promoter did not rescue the altered trichoblast patterning of *ccc1* roots (*Figure 3—figure supplement 1*). This is consistent with the *EXP7* promoter driving expression after cell fate determination, but before final cell length is achieved. Interestingly, expression of *GFP-CCC1* on the *EXP7* promoter did not rescue collet hair formation, which highlights the difference between root hair and collet hair development and indicates that different gene networks are important for collet hair formation compared to that of root hairs.

The stable expression of GFP-CCC1 in *Arabidopsis* revealed a similar subcellular localisation pattern to what we previously observed in *N. benthamiana* (*Henderson et al., 2015*), showing that CCC1 is localised to internal organelles in a native cell type. Time-lapse imaging of GFP-CCC1 movement was consistent with what could be expected for the Golgi or TGN/EE, however, GFP-CCC1 labelled organelles did not resemble the Golgi (*Videos 1 and 2*). To identify the GFP-CCC1 labelled compartments, we crossed the stably expressed marker, VHAa1-RFP, into plants expressing GFP-CCC1 (*Figure 3A*). Colocalisation of GFP-CCC1 and VHAa1-RFP was measured using object-based colocalisation analysis, with the ImageJ plugin DiAna, which revealed that 73% ± 9% of VHAa1-RFP colocalised with GFP-CCC1, while 58% ± 11% of GFP-CCC1 colocalised with VHAa1-RFP. The asymmetrical colocalisation indicates that, in addition to the TGN/EE, CCC1 might localise to additional organelles of the endomembrane system (*Figure 3A*), which is similar to NHX5 and NHX6 (*Reguera et al., 2015*). In addition, the Pearson's correlation coefficient of pixel signal intensity for RFP and GFP channels was calculated, which gave a correlation value of 0.86 ± 0.055. Pharmacological treatment further confirmed the endosomal localisation. Treatment with the trafficking inhibitor, brefeldin A (BFA), caused the GFP signal to accumulate in the centre of BFA bodies, consistent with a TGN/EE localisation (*Figure 3—figure supplement 1*).

We then investigated if CCC1 might cycle between the TGN/EE and PM. Cases like this have been observed for other ion transporters that localise mainly in endosomes but function at the PM, such as the iron transporter, IRT1. PM localisation of IRT1 can be visualised using pharmacological treatment with tyrphostin A23 (TyrA23), which acidifies the cytosol resulting in, among other effects, an inhibition in endocytosis and subsequent accumulation of cycling proteins at the PM (*Barberon et al., 2011*; *Dejonghe et al., 2016*). For CCC1, no change in the subcellular localisation was observed after treatment

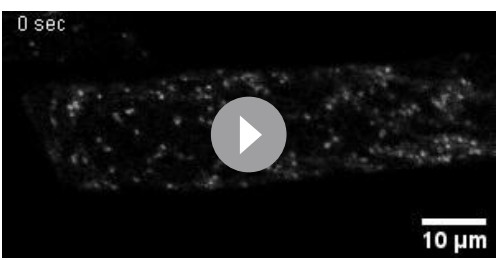

**Video 2.** Time-lapse showing movement of GFP-CCC1 labelled subcellular compartments in a root epidermal cell. GFP-CCC1 localises to motile intracellular organelles, expression driven with *EXP7* promoter. Time series was imaged below the radial plasma membrane (PM). 20 s shown per second at 10 frames per second.

https://elifesciences.org/articles/70701/figures#video2

with TyrA23 (*Figure 3—figure supplement 1*). Similarly, an osmotic shock treatment, which can sometimes induce a change in protein localisation (*Boursiac et al., 2008*; *Hachez et al., 2014*), did not lead to any observable changes in GFP-CCC1 localisation (*Figure 3—figure supplement 1*).

Loss or knockdown of other TGN/EE-localised proteins, such as the H⁺-V-ATPase, can affect the TGN/EE morphology (*Dettmer et al., 2006*). Transmission electron microscopy (TEM) of high-pressure-frozen, freeze-substituted samples revealed that the lack of CCC1 did not lead to obvious morphological changes in the TGN/EE ultrastructure, with the appearance of all organelles being similar between *ccc1* mutant and wildtype cells (*Figure 3—figure supplement 1*). This suggests that the defects observed in *ccc1* knockouts do not manifest through gross morphological changes in organelle structure, and so may instead be connected to other properties of the TGN/EE such as their luminal conditions.

## Hyperosmotic stress rescues root hair elongation and improves seed germination of ccc1

Knockouts of the TGN/EE-localised ion transporters, NHX5 and NHX6, as well as the proton pump V-H⁺-ATPase are salt-sensitive, with salt-associated germination and root growth phenotypes (*Krebs et al., 2010*; *Bassil et al., 2011*). We therefore further investigated the response of *ccc1* plants to salt stress. As rice OsCCC1.1 is implicated in maintaining cell sap osmolality (*Chen et al., 2016*), we also investigated the response to osmotic stress.

Plants were grown on media with added salt (NaCl), or an isosmotic concentration of mannitol or sorbitol, to determine if any changes in tolerance were the result of ionic or osmotic stress (*Figure 4A–B* and *Figure 4—figure supplement 1*). Wildtype and *ccc1* seeds were germinated on media containing 0–300 mM NaCl or 0–600 mM of mannitol or sorbitol. *ccc1* seeds typically germinated slightly earlier than wildtype seeds, so germination of seeds was assessed 8 days after imbibition (=6 days after transfer to the growth chamber). Under control conditions, over 90% of wildtype, *ccc1-1* and *ccc1-2* seeds germinated. The germination rate of all genotypes decreased with higher salt, mannitol and sorbitol concentrations, yet, this decrease was smaller in the *ccc1* knockouts than wildtype. On 450 mM mannitol, 30% ± 7% (SEM) of wildtype seeds germinated, while 79% ± 6% of *ccc1-1* and 64% ± 7% of *ccc1-2* seeds germinated. On 600 mM mannitol, 4% ± 2% of the wildtype seeds germinated, while 27% ± 6% of *ccc1-1* and 20% ± 6% of *ccc1-2* seeds were able to germinate in this condition. Similar results were obtained on sorbitol (*Figure 4—figure supplement 1*). A similar trend was also observable when plants were grown on media containing NaCl. Growth and germination of both wildtype and *ccc1* plants on NaCl was lower than observed on isosmotic concentrations of mannitol and sorbitol. In addition, the increased tolerance of *ccc1* seed germination to NaCl was less pronounced than on mannitol and sorbitol. At 150 mM NaCl, 67% ± 7% of wildtype seeds germinated compared with 85% ± 5% of *ccc1-1* and 95% ± 2% of *ccc1-2* seeds.

In addition to the higher germination rate of *ccc1* seeds on media with a high osmolarity, we observed that a striking phenotype of *ccc1* – the absence of collet hair formation (*Figure 2*) – was rescued when *ccc1* was grown on media with a high osmolarity. Under control conditions, over 90% of wildtype seedlings had a complete ring of collet hairs, while they were present in 0% of *ccc1* seedlings of either knockout allele (*Figure 4C–F* and *Figure 4—figure supplement 1*). When grown on 150 mM mannitol, 86% of *ccc1-1* seedlings and 75% of *ccc1-2* seedlings had collet hairs present as either a partial or complete ring around the hypocotyl. This improved to 100% of *ccc1-1%* and 98% of *ccc1-2* on media with 300 mM mannitol. As occurred for germination, little difference was seen between mannitol and sorbitol treatments (*Figure 4—figure supplement 1*). NaCl treatment also resulted in the presence of collet hairs in *ccc1* knockouts, however, not to the same magnitude as observed in mannitol and sorbitol treatments. When treated with 150 mM NaCl, 98% of *ccc1-1%* and 63% of *ccc1-2* had collet hairs. Inspection of the root-hypocotyl base region where collet hairs form under control conditions shows that collet hairs do initiate in *ccc1* plants but fail to elongate (*Figure 4—figure supplement 1*). This suggests that the absence of collet hairs in *ccc1* is due to cell elongation defects, with no obvious defects in cell identify (*Figure 4—figure supplement 1*). This further suggests that osmotic stress rescues cell elongation defects in *ccc1* roots.

To further investigate if raising external osmolarity leads to increased cell elongation in *ccc1* roots, we investigated if the reduced speed of root hair growth of *ccc1* plants is also rescued by increasing external osmolarity. To determine root hair elongation speed, root hairs below a final length of 50 μm

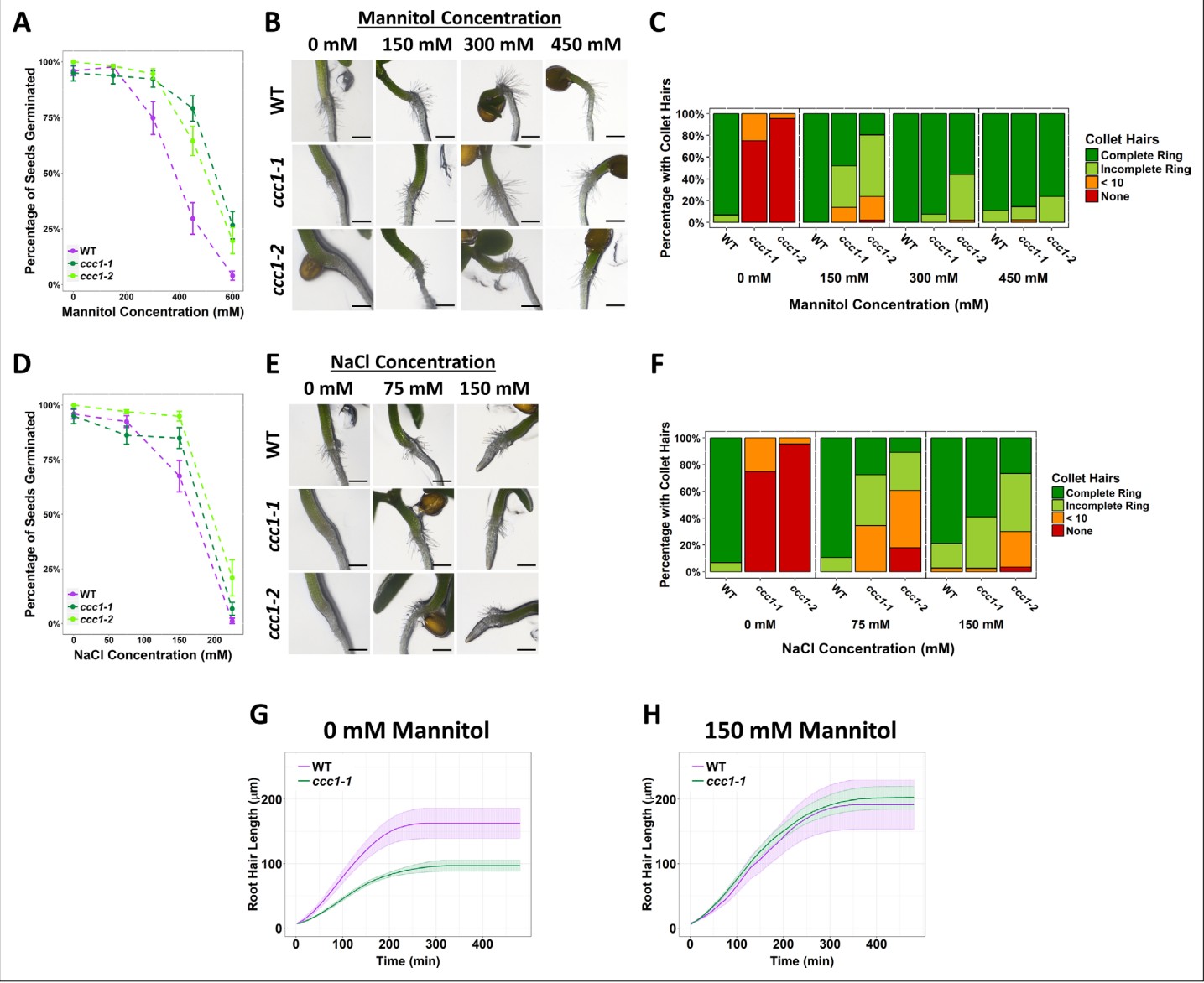

**Figure 4.** Increased external osmolarity rescues cell elongation defects in *ccc1*. (**A, D**) A higher percentage of *ccc1* seeds germinate on media with a higher osmolarity, adjusted with increasing (A) mannitol or (D) NaCl concentrations. Number of germinated seeds assessed 6 days after imbibition. n > 90 seeds. (**B–C, E–F**) Collet hair formation in *ccc1* is rescued on media with a higher osmolarity. n > 9 plants for 450 mM treatments, >30 plants for all other treatments. Scale bars are 300 µm. Germination and collet hair assays with mannitol, sorbitol, and NaCl were performed together and therefore share the same control (0 mM). (**G–H**) The slower rate of root hair elongation in *ccc1* is rescued when grown in media with 150 mM mannitol (see *Videos 3–6*). Under control conditions, between 50 and 100 min after initiation, wildtype roots hairs elongated at a rate of 0.88 ± 0.27 µm min⁻¹ compared with a rate of 0.47 ± 0.08 µm min⁻¹ in *ccc1*.

The online version of this article includes the following source data and figure supplement(s) for figure 4:

**Source data 1.** Data analysed and presented in *Figure 4* and supplement.

**Figure supplement 1.** Germination of *ccc1* seeds is tolerant to osmotic stress imposed by sorbitol and *ccc1* root hair elongation is rescued on sorbitol.

were not measured, this consists of all root hairs that initiated, but did not elongate. We did this, as it was not possible to acquire an accurate measurement of elongation rate from such root hairs. Time-lapse microscopy revealed that *ccc1* root hairs grew for the same duration as wildtype hairs, but at a reduced speed leading to a reduced final root hair length (*Figure 4G*, *Videos 3–6*). Between 50 and 100 min after elongation initiated, wildtype root hairs had an average elongation rate of 0.88 ± 0.27 µm min⁻¹, while *ccc1* root hairs elongated at half that speed, at 0.47 ± 0.08 µm min⁻¹ (*Figure 4G*). Strikingly, when grown in media with 150 mM mannitol, the elongation rate of *ccc1* root

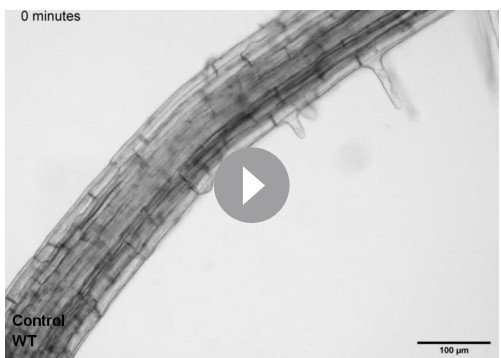

**Video 3.** Wildtype control.Time-lapse of root hair elongation. 48 min shown per second at 24 frames per second. See also Figure 4G.

https://elifesciences.org/articles/70701/figures#video3

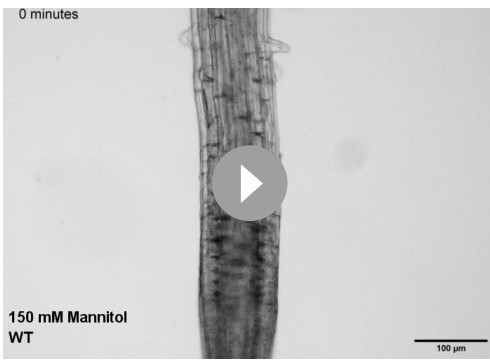

**Video 5.** Wildtype 150 mM mannitol.Time-lapse of root hair elongation.
48 min shown per second at 24 frames per second. See also Figure 4H.

https://elifesciences.org/articles/70701/figures#video5

hairs was increased to be the same rate as wild-type (*Figure 4H*). Final length of *ccc1* root hairs was also rescued in solutions with high external osmolarity.

## ccc1 *cells do not have higher cell sap osmolality but are more resistant to plasmolysis*

We next investigated the origin of rescue of *ccc1* root hair elongation when grown on media with an elevated osmolarity. Conceivably, this could be connected to an increased osmolarity in *ccc1* cells. However, we found that cell sap osmolality was not higher in *ccc1* compared to wildtype (*Figure 5A*).

Other factors that could contribute to the improved growth of *ccc1* plants on media with high osmotic strength include altered osmoregulatory capacity. To explore if the osmoregulatory capacity of *ccc1* cells may be altered, the point of incipient plasmolysis was determined for wildtype and *ccc1* epidermal root cells. This determines the external osmotic strength required to induce plasmolysis in 50% of the cells. Plasmolysis of mature epidermal root cells was induced by submerging roots in liquid media containing high concentrations of mannitol for 30 min after which plamolysed cells were observed using plants that stably express a PM marker. A lower proportion of plasmolysed cells were counted in *ccc1* at all but the highest mannitol concentration (*Figure 5B–C*). The greatest difference was observed at 250 mM mannitol, where 71% ± 9% (SEM) of wildtype cells were plasmolysed compared to only 23% ± 9% of *ccc1* cells at the same concentration. The estimated mannitol concentration for incipient plasmolysis was determined

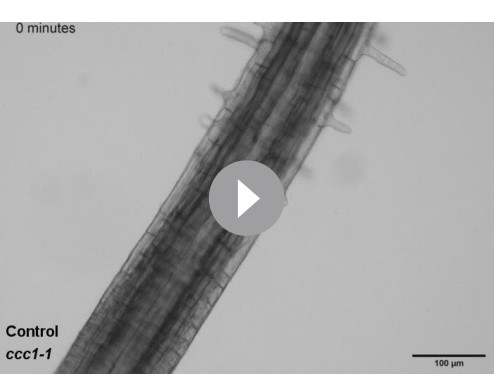

**Video 4.** *ccc1* control Time-lapse of root hair elongation. *ccc1* root hairs elongate slower than wildtype under control conditions. 48 min shown per second at 24 frames per second. See also Figure 4G.

https://elifesciences.org/articles/70701/figures#video4

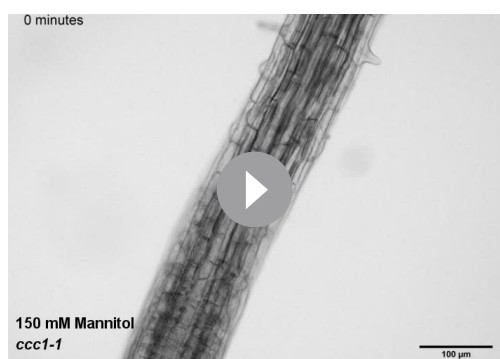

**Video 6.** *ccc1* 150 mM mannitol.Time-lapse of root hair elongation. While *ccc1* root hairs elongate slower than wildtype under control conditions they elongate at the same speed as wildtype when grown on media containing 150 mM mannitol. 48 min shown per second at 24 frames per second. See also Figure 4.

https://elifesciences.org/articles/70701/figures#video6

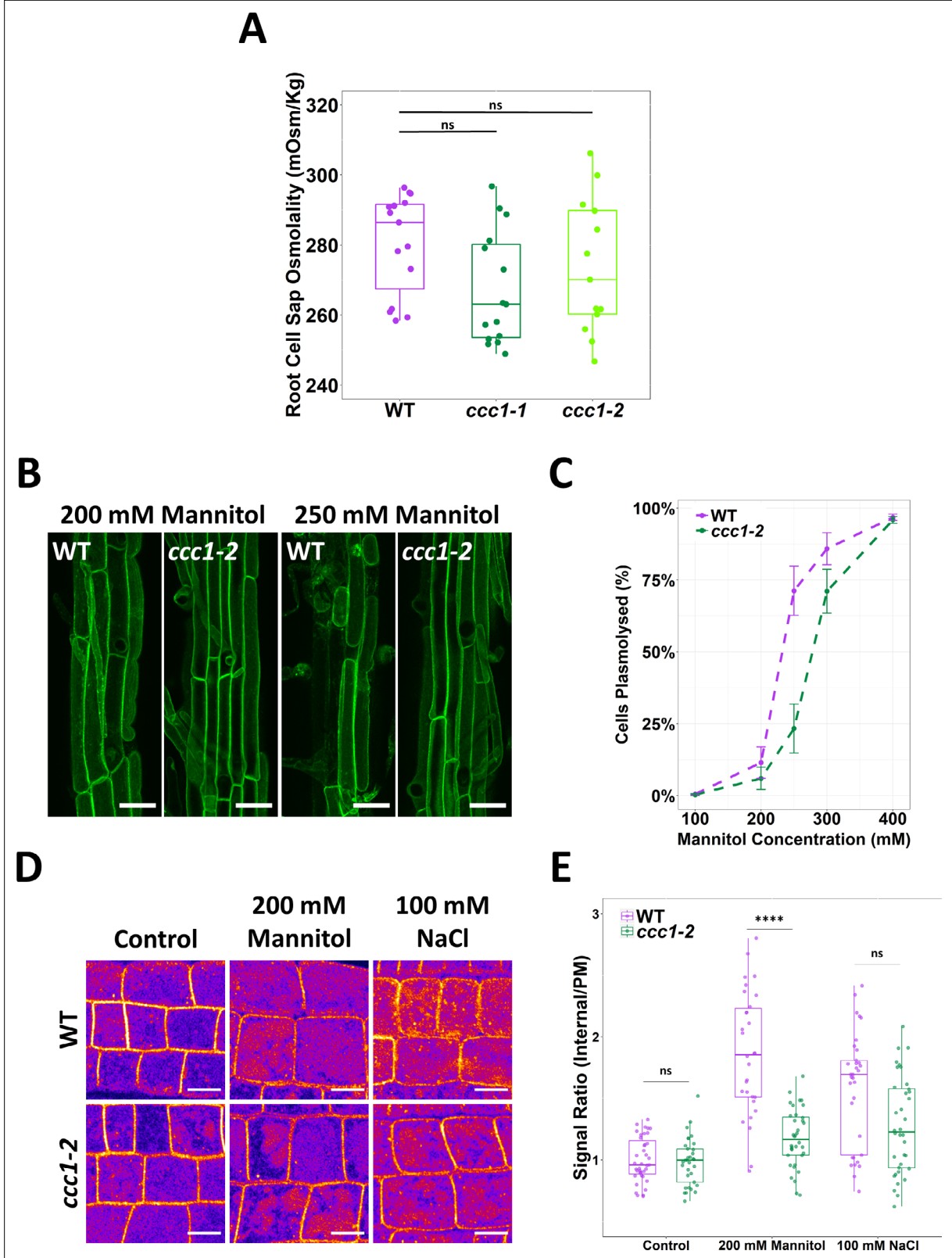

**Figure 5.** *ccc1* root cells do not have a higher osmolality but are more tolerant to plasmolysis. (**A**) The cell sap osmolality of whole root fluid was measured using a freeze point osmometer; no difference between wildtype and *ccc1* cell sap osmolality was found. n = 13–15. (**B–C**) *ccc1* root epidermal cells require a higher external mannitol concentration to induce incipient plasmolysis. Plasmolysed and non-plasmolysed cells were counted 1 hr after treatment. The plasma membrane (PM) of cells was visualised with the PM marker, GFP-LIT6b (green). Forty cells were assessed per plant, 12 plants were

*Figure 5 continued on next page*

Figure 5 continued

counted for each genotype/treatment. Scale bars are 40 μM. (**D–E**) The internalisation of PIP2;1-GFP in response to osmotic (200 mM mannitol) and salt (100 mM NaCl) shock treatment was assayed in root epidermal cells of the elongation zone. *ccc1* has a reduced PIP2;1-GFP internalisation after osmotic shock as compared to wildtype. Plants were imaged 30 min after treatment. A signal ratio of internal and PM signal was used to measure internalisation. Data presented relative to wildtype control. n > 45. Scale bars are 10 μM. Boxplot shows range; median, first and third quartile are indicated. One-way ANOVA used to determine p. **** indicates $p < 0.0001$.

The online version of this article includes the following source data for figure 5:

**Source data 1.** Data analysed and presented in *Figure 5*.

using a fitted curve. For wildtype, 50% of cells were estimated to plasmolyse at 232 mM mannitol, while plasmolysis of 50% of *ccc1* root cells was estimated to require a mannitol concentration of 277 mM, 45 mM higher than that of the wildtype. As a higher cell sap osmolality is excluded as a reason for higher resistance to plasmolysis in *ccc1*, other factors are the cause for the observed differences between the wildtype and *ccc1*.

## Trafficking of PIP2;1 in response to osmotic shock is reduced

In response to osmotic stress, aquaporins are removed from the PM (*Boursiac et al., 2008*; *Hachez et al., 2014*). The aquaporin PIP2;1 cycles between the PM and TGN/EE and is internalised in response to osmotic stress (*Luu et al., 2012*). We investigated osmotically induced internalisation of stably expressed PIP2;1-GFP in *ccc1* and wildtype plants. Wildtype and *ccc1* roots were treated with 100 mM NaCl or 200 mM mannitol for 30 min before internalisation of PIP2;1-GFP was measured by obtaining the ratio of the fluorescent signal on the PM compared to the signal inside the cell. In both wildtype and *ccc1* roots, PIP2;1-GFP remained localised to the PM when treated with only 1/2 MS (*Figure 5D–E*). When treated with NaCl or mannitol, both genotypes internalised PIP2;1-GFP, but the relative portion of PIP2;1-GFP internalisation was lower in *ccc1* than wildtype (*Figure 5D–E*). This excludes a more efficient removal of PIP2;1 from the PM as the reason for the reduced plasmolysis in *ccc1* knockouts. Additional factors, such as the previously observed alterations in cell wall composition in *ccc1* (*Han et al., 2020*), or changes in cellular signalling connected to ion transport at the TGN/EE might therefore contribute to observed improvement of the *ccc1* knockout when grown in media with higher external osmolarity.

## Loss of CCC1 results in defects of endomembrane trafficking

The reduced internalisation of PIP2;1-GFP in *ccc1* cells in response to osmotic stress may be the result of a reduced rate of endomembrane trafficking in *ccc1* cells. We therefore investigated if *ccc1* is important for maintaining normal rates of endomembrane trafficking.

The rate of protein exocytosis, which is a combination of trafficking from secretion and recycling, was assayed using wildtype and *ccc1* lines stably expressing PIN2-GFP. BFA treatment resulted in the accumulation of PIN2-GFP in the endomembrane system, measured as a strong increase in cytoplasm:PM signal ratio. This increase was more pronounced in *ccc1* than in wildtype cells (*Figure 6A–B*). BFA treatment in *Arabidopsis* root cells leads to the formation of BFA bodies, which are an amalgamation of Golgi and TGN/EE (*Geldner et al., 2001*). Interestingly, while BFA bodies were compact and bright in wildtype plants, BFA bodies in *ccc1* were less dense with a more diffuse GFP signal pattern. Upon BFA washout, wildtype plants exhibited an almost complete recovery of the PIN2-GFP cytoplasm:PM signal ratio, suggesting that protein secretion and recycling had resumed. In contrast, *ccc1* showed minimal recovery in the cytoplasm:PM signal ratio indicating a considerable decrease in PIN2-GFP exocytosis (*Figure 6A–B*). PM fluorescence of PIN2 is lower in other mutants with loss of TGN/EE ion transporters, such as *nhx5/nhx6* plants (*Dragwidge et al., 2019*). Quantification of PM fluorescence of PIN2 in the *ccc1* plants revealed that PIN2 signal intensity at the PM was lower in *ccc1* (*Figure 6—figure supplement 1A*), suggesting that the loss of this TGN/EE-localised ion symporter may be important for PM accumulation of proteins like PIN2. As there is a difference in fluorescence of PIN2 between wildtype and *ccc1* lines, we checked that a difference in overall fluorescence did not influence the fluorescence ratio being used to measure internalisation. No correlation was found between total fluorescence and ratio within a single treatment and genotype suggesting that total fluorescence does not affect the ratio (*Figure 6—figure supplement 1B*).

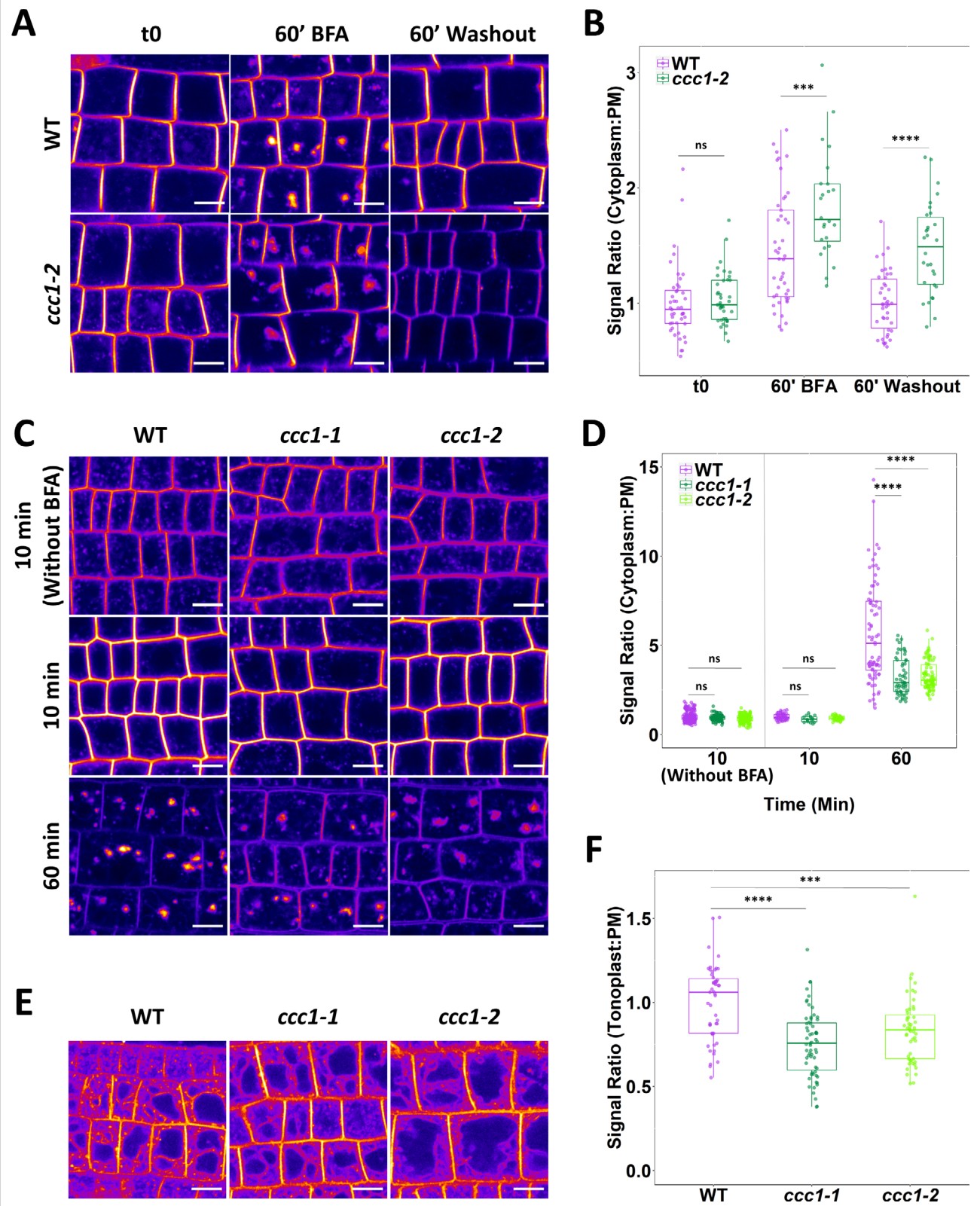

**Figure 6.** Loss of CCC1 leads to defects in exocytosis and endocytic trafficking. (**A–B**) *ccc1* root epidermal cells show a reduced recovery of PIN2-GFP cytoplasm:PM signal ratio after a 60 min treatment with 25 µM brefeldin A (BFA). n > 24 cells, 3 cells measured per plant. (**C–D**) Endocytic trafficking of the membrane dye FM4-64 (red), measured as an increase in the cytoplasm:PM ratio, is reduced in *ccc1* lines while endocytosis, measured with a separate set of experiments without the use of BFA, was unchanged in *ccc1*. Plants were kept in 25 µM BFA for the duration of the experiment. n

*Figure 6 continued on next page*

*Figure 6 continued*

> 56 cells of >9 plants. (**E–F**) Trafficking of FM4-64 to the vacuole in the absence of BFA, measured as an increase of the tonoplast:PM signal ratio, is reduced in ccc1 lines. n > 51 cells of >18 plants. Boxplots show range; median, first and third quartile are indicated. Points represent individual measurements. All images are single representative optical sections. All data presented relative to wildtype under control conditions. One-way ANOVA used to determine p. *** indicates $p < 0.001$, **** indicates $p < 0.0001$. All scale bars are 10 µm.

The online version of this article includes the following source data and figure supplement(s) for figure 6:

**Source data 1.** Data analysed and presented in *Figure 6* and supplement.

**Figure supplement 1.** PIN2-GFP fluorescence on the plasma membrane (PM) is lower in *ccc1* and trafficking defects in *ccc1* are not pre-trans-Golgi-network/early endosome (TGN/EE).

As the rate of PIP2;1-GFP internalisation was reduced in *ccc1*, we also measured the rate of endocytic trafficking. Endocytic trafficking was quantified by internalisation of the endocytic tracer FM4-64, measured by an increase in the cytoplasm:PM signal ratio with BFA treatment. After 60 min of treatment (10 min with FM4-64 and BFA, 50 min with BFA only), the fluorescence ratio was much lower in *ccc1* than wildtype cells, consistent with a significant reduction in the rate of endocytic trafficking (*Figure 6C–D*). We additionally investigated if endocytosis was disrupted in *ccc1* mutants through a pulse-chase experiment without BFA treatment. No difference in internalisation of the FM4-64 was observed after 10 min (*Figure 6C–D*), indicating that endocytic trafficking, but not endocytosis itself, is affected in *ccc1*. In a third FM4-64 pulse-chase experiment, trafficking of FM4-64 to the vacuole was measured without the use of BFA, and the ratio of FM4-64 fluorescence at the tonoplast and the PM was compared. Labelling of the tonoplast with FM4-64 became visible in both wildtype and mutant cells after 3 hr, with a reduced ratio of tonoplast:PM signal in *ccc1* compared with wildtype cells. This indicates a reduction in endocytic trafficking of FM4-64 to the vacuole when CCC1 is absent (*Figure 6E–F*).

In both experiments using BFA, BFA bodies in *ccc1* appeared to have a more diffuse morphology with a duller signal compared to the more compact and therefore brighter appearing BFA bodies in the wildtype (*Figure 6A and C*). This possible difference in BFA body formation had, however, no impact on the cytoplasm:PM signal ratio measurements, as for the experiment using PIN2 we found a higher ratio in *ccc1*, and for the experiment using FM4-64 we found a lower ratio. Our combined results therefore indicate general defects in exocytosis and endocytic trafficking when CCC1 function is lost.

## CCC1 contributes to TGN/EE luminal pH regulation

Here, we show that GFP-CCC1 is localised in the TGN/EE and identified that CCC1 is important for endomembrane trafficking (*Figures 3 and 6*). We hypothesised that the symporter might impact endomembrane trafficking through its role in regulating ion movement across the TGN/EE membrane, which may contribute to an altered luminal pH by affecting the activities of the cation- and anion-proton exchangers, and therefore ultimately the proton pump. To investigate this possible role of CCC1 in the pH regulatory network, we used both pharmacological treatments and a stably expressed TGN/EE fluorescent pH sensor.

The ionophore monensin was utilised to assess if the loss of CCC1 results in changes to the ability of the TGN/EE to maintain a stable luminal environment. Monensin is a monovalent cation ionophore, acting as a membrane permeable ion exchanger, exchanging luminal protons for cations (*Zhang et al., 1993*). The rapid increase in osmotically active cations within the TGN/EE lumen caused by monensin leads to observable TGN/EE swelling. The cation concentration gradient is an important factor for determining the severity of monensin-induced TGN/EE swelling. This is exemplified by *nhx5/nhx6* mutants, which are less sensitive to monensin despite a more acidic luminal pH and greater proton gradient for driving swelling (*Dragwidge et al., 2019*). We hypothesised that if CCC1 is important for ion efflux out of the TGN/EE, *ccc1* TGN/EE might be hypersensitive to monensin-induced TGN/EE swelling. We assessed the susceptibility of *ccc1* to monensin using live-cell imaging of VHAa1-RFP-labelled TGN/EE in wildtype and *ccc1* root epidermal cells in the elongation zone, treated with 2.5 µM of monensin for 15 min. The average wildtype TGN/EE size increased by 28% ± 13% after treatment, while *ccc1* TGN/EE size increased by 52% ± 16% (*Figure 7A*), revealing a highly increased susceptibility of *ccc1* TGN/EE to osmotically induced swelling caused by cation accumulation. This suggests that CCC1 mitigates the effect of monensin in wildtype cells, by allowing efflux of $K^+$ and $Cl^-$, as anion

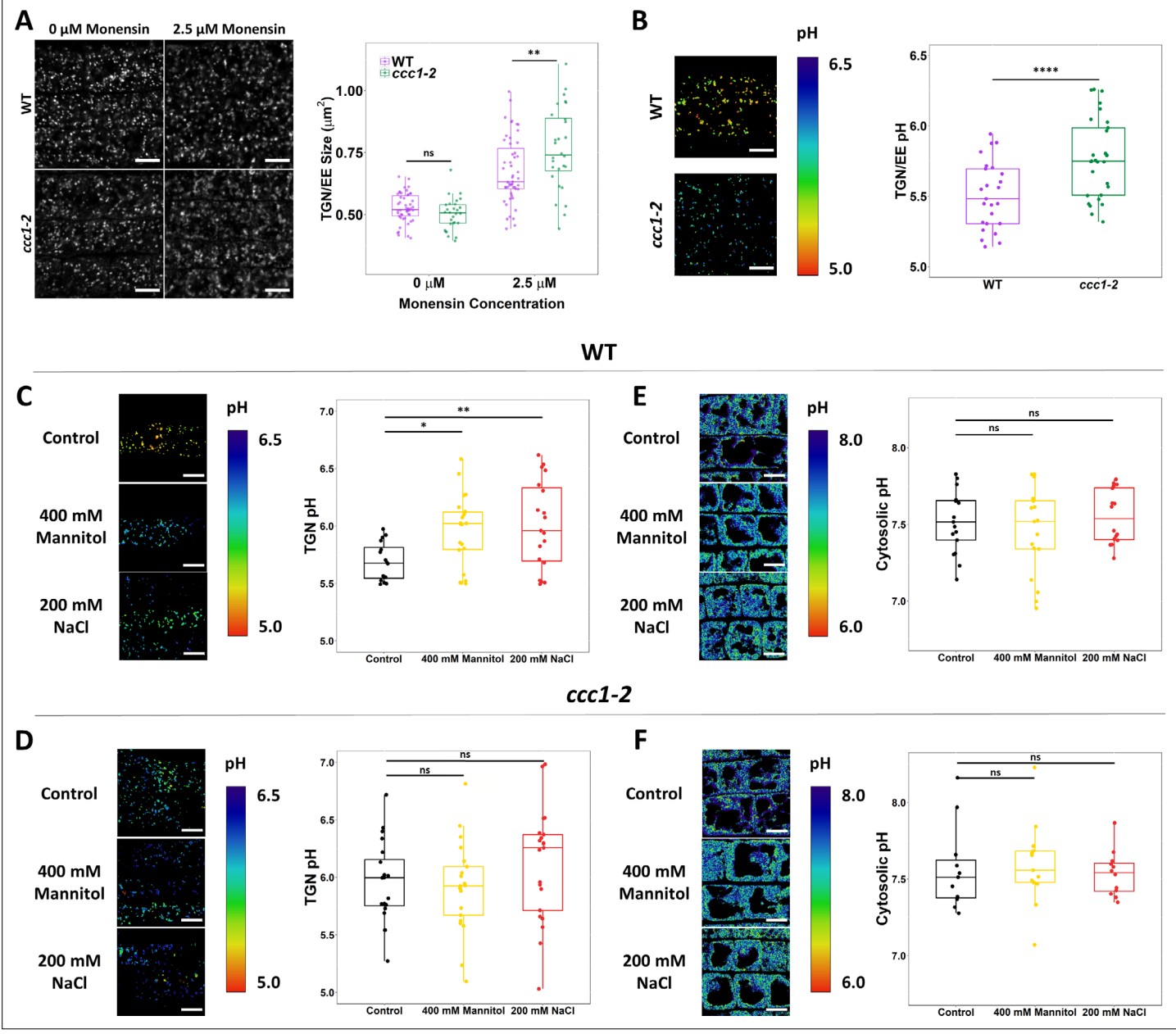

**Figure 7.** CCC1 is required for regulation of trans-Golgi-network/early endosome (TGN/EE) luminal conditions and luminal pH change in response to osmotic shock. (**A**) Osmotically induced TGN/EE swelling due to cation influx is more severe in *ccc1* epidermal root cells after 15 min of treatment with 2.5 µM monensin. TGN/EE are visualised through stable expression of VHAa1-RFP (white) in wildtype and *ccc1* backgrounds. n > 28 plants. Scale bars are 10 µm. Images are single representative optical sections. (**B–D**) Stable expression of the TGN/EE lumen pH sensor pHusion in wildtype and *ccc1* backgrounds. (**C**) The pH of *ccc1* TGN/EE is 0.3 pH units higher compared to wildtype in root epidermal cells. n > 26. (**D**) The pH of wildtype TGN/EE increased by 0.3 in response to 15 min treatments with 400 mM mannitol and 200 mM NaCl. n > 16. (**E**) The same treatments resulted in no change to the pH of *ccc1* TGN/EE. n > 16. (**E–F**) Stable expression of the cytosolic pH sensor pHGFP in wildtype and *ccc1* backgrounds. There is no significant pH difference between the two genotypes in root epidermal cells. The cytosolic pH of both (**E**) wildtype and (**F**) *ccc1* was unchanged in response to osmotic and salt shock. n > 13. pH with both sensors was calculated using calibration curves performed with each set of experiments. Images show the ratio of GFP/RFP of the pH sensor SYP61-pHusion in the TGN/EE lumen and emission ratio for the pHGFP cytosolic sensor as a colour range, using the inverted 'physics' look-up table in ImageJ. Scale bars are 10 µm. Calibration curves for pH sensors displayed in *Figure 7—figure supplement 2*. Boxplots show range; median, first and third quartile are indicated. Points represent individual measurements. One-way ANOVA used to determine p. * indicates p < 0.05, ** indicates p < 0.01, **** indicates p < 0.0001.

The online version of this article includes the following source data and figure supplement(s) for figure 7:

**Source data 1.** Data analysed and presented in *Figure 7* and supplements.

*Figure 7 continued on next page*

**Figure supplement 1.** Loss of CCC1 does not impact vacuolar pH.

**Figure supplement 2.** Calibration curves of SYP61-pHusion, pHGFP, and BCECF-AM.

**Figure supplement 3.** *ccc1/nhx5/nhx6* produce no viable seeds.

and cation transport are stochiometrically linked in CCC1. We further suspected that TGN/EE pH could be altered as a consequence of altered cation and anion transport processes.

To investigate this hypothesis, we introduced the stably expressed TGN/EE-localised pH sensor, SYP61-pHusion (*Luo et al., 2015*), into *ccc1* and investigated if defects in cation anion symport can impact pH regulation in the TGN/EE. Confocal imaging of epidermal cells in the root elongation zone revealed that luminal pH was more alkaline in *ccc1*, with a pH of 5.8 ± 0.05, compared to wildtype pH of 5.5 ± 0.05, confirming our initial hypothesis that a non-proton transporting ion symporter can impact luminal pH (*Figure 7B*). We investigated whether loss of CCC1 led to a more general defect of intracellular pH by measuring cytosolic pH using the genetically encoded and stably expressed pHGFP (*Moseyko and Feldman, 2001*) and vacuolar pH using the pH-sensitive dye, BCECF (*Krebs et al., 2010*). No difference was found between the *ccc1* and wildtype, demonstrating that lack of CCC1 leads to spatially defined pH changes instead of a general effect (*Figure 7E–F*, *Figure 7— figure supplement 1*). To further assay the role of CCC1 in the TGN/EE and how it may interact with other TGN/EE transporters and the proton pump, crosses were made of *ccc1 × det3* and *ccc1 × nhx5 × nhx6*. However, crosses either resulted in plants which were not viable or no combined mutant could be obtained (*Supplementary file 1b*, *Figure 7—figure supplement 3*). This might further support that CCC1 is required for the regulation of the TGN/EE luminal conditions in concert with other ion and proton transporters, and that removal of more regulatory components leads to such severe defects that the plants are unable to survive.

## The luminal pH of the TGN/EE increases in response to salt and osmotic stress

Plants with altered TGN/EE lumen pH, including *ccc1* plants, show altered growth responses to salt and osmotic treatments. Therefore, we investigated in wildtype if there are changes to TGN/EE luminal pH in response to salt and osmotic treatment and then compared our findings to *ccc1*.

The TGN/EE luminal pH of epidermal root cells of the elongation zone was quantified after 15 min exposure to 200 mM NaCl or 400 mM mannitol and compared to that of roots exposed to 1/2 MS control solution. In wildtype, in response to both NaCl and mannitol, the pH of the TGN/EE increased from 5.7 ± 0.04 (SEM) under control conditions to 6.0 ± 0.09 and 6.0 ± 0.07, respectively (*Figure 7C*). Mannitol and NaCl both elicited the same pH response from plants demonstrating that the pH change is not exclusively connected to an increase in external ion concentrations but rather occurs under an elevated external osmolarity. Osmotic stress triggers an array of responses at the PM that can result in membrane depolarisation and depletion of cellular ATP (*Che-Othman et al., 2017*). Such events may impact proton transport across multiple membranes and as such, pH changes under osmotic stress may not be exclusive to the TGN/EE lumen. To assess this, the pH of the cytosol was measured in response to stress. We excluding the signal from nuclei for measurements. This was done because the pH of the nucleus differs from that of the cytosol and its response to stress may also differ (*Shen et al., 2013*). We found that neither 15 min NaCl nor mannitol treatment induced significant changes in cytosolic pH (*Figure 7E*). Under control conditions, the pH of the wildtype cytosol was 7.5 ± 0.05, almost identical pH values of 7.5 ± 0.07 under NaCl treatment and 7.6 ± 0.04 under mannitol treatment were obtained. This suggests that immediate cellular signalling in response to salt and osmotic shock does not involve changes in cytosolic pH but changes in luminal TGN/EE pH.

In the same set of experiments, the TGN/EE and cytosolic pH of *ccc1* mutants in response to salt and osmotic shock treatments was measured. As in wildtype, no changes were observed for *ccc1* in cytosolic pH response; *ccc1* cytosolic pH under control conditions was 7.5 ± 0.12, in NaCl it was 7.6 ± 0.08, and was 7.5 ± 0.04 when treated with mannitol (*Figure 7F*). In addition, cytosolic pH in wildtype and *ccc1* under control conditions are almost identical, further supporting the idea that the pH change in *ccc1* is compartment specific and not a generalised effect. In agreement with our experimental data set presented in *Figure 7B*, the TGN/EE luminal pH of *ccc1* TGN/EE was higher than wildtype, with a

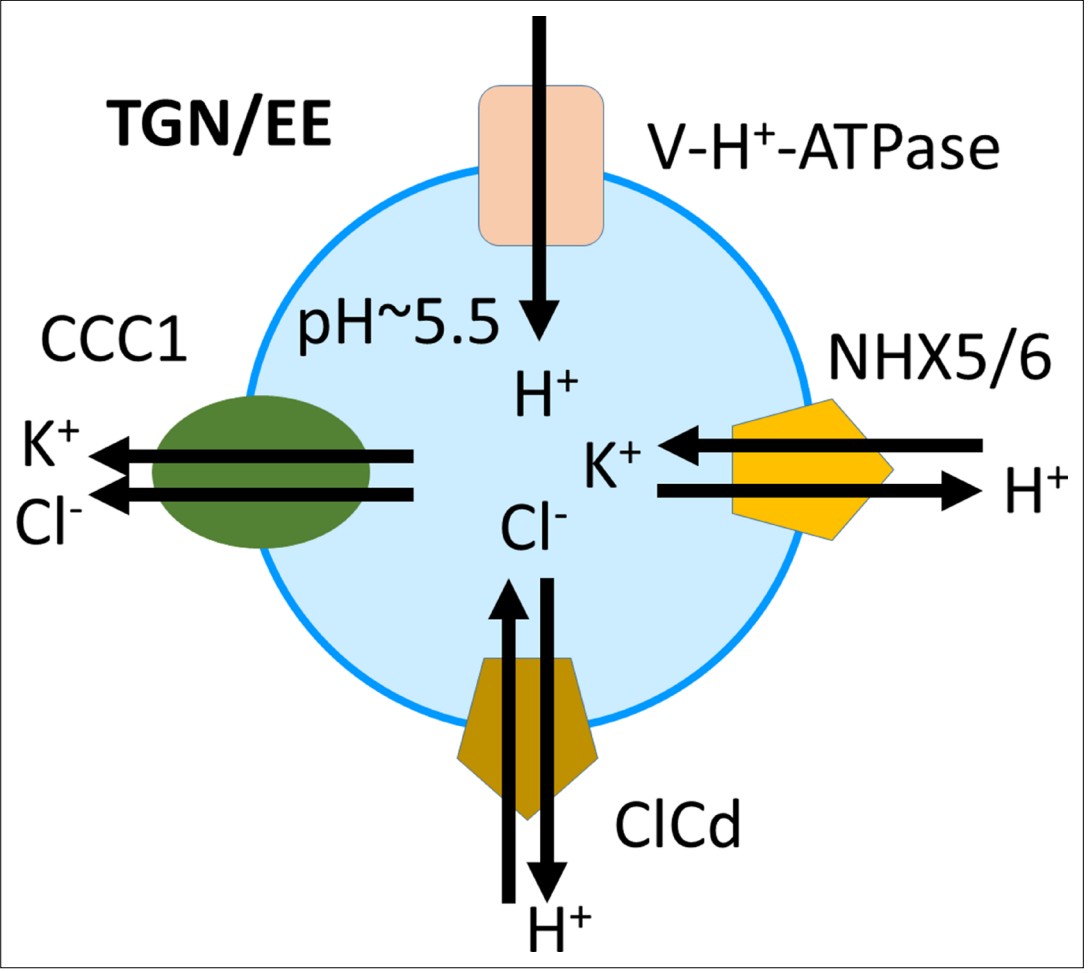

**Figure 8.** Proposed model of ion and pH regulation in the trans-Golgi-network/early endosome (TGN/EE). The V-H⁺-ATPase proton pump, the cation-proton exchangers NHX5 and NHX6, and the anion-proton exchanger CLCd are important for acidification and fine-tuning of the pH regulation in the TGN/EE lumen. CCC1 provides a cation and anion efflux mechanism, completing the regulatory transport circuit.

pH of 6.0 ± 0.08 in control conditions. Different from wildtype, no significant change in luminal pH was detected in *ccc1* knockouts with a mean TGN/EE pH of 6.1 ± 0.11 (SEM) and 5.9 ± 0.08 in response to NaCl and mannitol, respectively (*Figure 7D*).

Collectively, these results show that there is a TGN/EE-specific pH response to osmotic stress in wildtype plants. This cellular response was not found in *ccc1*, which might indicate that the TGN/EE luminal pH is set to 6.0 in response to stress or that pH 6 may be an upper limit for the TGN/EE pH. The unresponsiveness of TGN/EE luminal pH to osmotic shock in *ccc1* might be connected to its improved growth under high external osmolarity.

## Discussion

In the TGN/EE, both an increase or a decrease of luminal pH leads to changes in endomembrane trafficking, cell expansion, and cell wall formation, which implies that the pH of the TGN/EE is finely regulated (*Luo et al., 2015*; *Dragwidge et al., 2019*). Here, we show that a functional CCC1 is localised in the TGN/EE in *Arabidopsis*, and that loss of this cation anion symporter leads to defects in establishing the typical, low pH in TGN/EE lumen. This makes CCC1 the first TGN/EE-localised transporter important for pH regulation that has not been shown to directly transport protons. Our results show that CCC1 is important for counteracting the cation-induced osmotic swelling of TGN/EE by monensin (*Figure 7*), consistent with this symporter being important for regulating luminal ion

concentrations. pH measurements confirmed that *ccc1* cells have a more alkaline pH in the TGN/EE lumen compared to wildtype. We therefore propose to update the current model of pH regulation, containing the V-H⁺-ATPase, NHX5 and NHX6, and CLCd (*Sze and Chanroj, 2018*), adding CCC1 (*Figure 8*).

The function of CCC1 in the TGN/EE is consistent with CCC1 localisation to this organelle, as detected here and shown in earlier studies (*Sadowski et al., 2008*; *Drakakaki et al., 2012*; *Groen et al., 2014*; *Nikolovski et al., 2014*; *Henderson et al., 2015*; *Parsons et al., 2019*). We find no evidence that CCC1 was functional on any other membrane. The construction of a functional CCC1 with a tag was not trivial. Multiple attempts at tagging CCC1 failed and the construct used here, GFP-CCC1, did not give transformants if ubiquitously expressed under the control of its native promoter (*Supplementary file 1a*). We demonstrate cellular functionality through the ability of GFP-CCC1 to improve root hair and trichoblast elongation (*Figure 3* and *Figure 3—figure supplement 1*), yet, the inability to express tagged CCC1 ubiquitously suggests that tagging does affect CCC1, potentially through modifying regulation or activity of the protein to create an overactive, embryo lethal variant. In another study, a CCC1-GFP construct expressed in pollen was predominantly localised to endomembranes, but was also suggested to localise to the PM (*Domingos et al., 2019*). This might indicate that in pollen, CCC1 might have other cellular localisation in addition to TGN/EE. However, the functionality of this construct was not shown through complementation of the pollen tube growth defect, and the PM signal might have been a result of overexpression of the construct through the tomato-derived Lat52 promoter or the C-terminal GFP-tag may have perturbed retention of the CCC1-GFP construct in the TGN/EE. Our results in root cells demonstrate that functional GFP-CCC1 localises to the TGN/EE and that loss of function of the symporter impacts TGN/EE luminal conditions, and results in phenotypes in line with disruption in TGN/EE luminal pH.

Interestingly, endomembrane localisation of CCCs is not restricted to *Arabidopsis* CCC1. Yeast CCC, called VHC1, is the only functionally characterised CCC outside of plants and animals. Similar to *Arabidopsis* CCC1, VHC1 is found on an intracellular membrane; and like many animal CCCs, and CCC1 from plants, VHC1 is involved in osmoregulation (*Petrzselyova et al., 2013*). Intracellular organelles such as the TGN/EE (or the yeast vacuole) are kept in osmotic equilibrium with the cytosol, which is akin to the adjustment animal cells conduct with their extracellular fluid. Unlike animal cells, plants maintain steep osmotic gradients across their plasma membranes, enabled by the presence of a cell wall and the establishment of turgor pressure. Therefore, proteins involved in osmoregulation localised on the PM of animal or plant cells are likely to require different characteristics and this may explain the difference in localisation of CCC proteins in animals as compared to organisms with a cell wall. Interestingly, also for CLC proteins, many animal CLCs are localised to the plasma membrane, but all investigated plant CLCs are localised to internal membranes, with no PM localisation detected (*De Angeli et al., 2009*).

CCC1 is an electroneutral anion cation symporter, which, like other CCCs, fixes the stoichiometry of exported anions and cations at 1:1 (*Colmenero-Flores et al., 2007*). In animals, this stoichiometry either consists of the electroneutral transport of 2Cl⁻:1Na⁺:1K⁺ in NKCCs, or 1Cl⁻:1K⁺ and 1Cl⁻:1Na⁺ in KCCs and NCCs, respectively. As *Arabidopsis* plants can be grown in the near absence of Na⁺ (50 nM) without showing a *ccc1* knockout phenotype, it is likely that CCC1 is functional in the absence of Na⁺. This supports the hypothesis that CCC1 can function as a 1Cl⁻:1K⁺ symporter, which could be validated through further experimentation in heterologous systems (e.g. *Colmenero-Flores et al., 2007*; *Henderson et al., 2015*; *van Delden et al., 2020*).

In the TGN/EE, a 1:1 K⁺:Cl⁻ ion export ratio of plant CCCs would tightly connect the activities of the colocalised ion exchangers NHX5 and NHX6 with CLCd, which contribute to the strict but dynamic regulation of pH. In addition, other transporters such as the potassium efflux antiporters KEA4, KEA5, and KEA6 might also contribute to luminal pH regulation in planta; yet their roles are currently less understood since triple *kea4/kea5/kea5* knockouts show a much less severe phenotype compared to *nhx5/nhx6*, *det3*, or *ccc1* (*Dettmer et al., 2006*; *Colmenero-Flores et al., 2007*; *Bassil et al., 2011*; *Zhu et al., 2019*).

Functional pH regulation and endomembrane transport has a vital role in many cellular processes, including cell wall formation, nutrient acquisition, and the establishment of hormone gradients (*Guo et al., 2014*; *Adamowski and Friml, 2015*; *Sinclair et al., 2018*). In agreement with this, *ccc1* mutant plants display very severe phenotypic alterations compared to wildtype. We found that CCC1 is

ubiquitously expressed, and its function is required for cell elongation, which might be a contributing cause of the reduced overall growth of the mutants (*Figures 1 and 2*). Cell elongation is also reduced in *nhx5/nhx6* and *clcd/amiR-clcf* mutants; with reduced root hair elongation speed in both and reduced root epidermal cell and root hair length in *clcd/amiR-clcf* (*Bassil et al., 2011*; *Scholl et al., 2021*). Reduced root hair elongation could be the result of reduced rates of endomembrane trafficking, which have been observed in *nhx5/nhx6* causing reduced delivery of cargo important for cell expansion such as cell wall components. Phenotypic defects in *ccc1* are not restricted to cell elongation, we found that trichoblast patterning was also altered in *ccc1* roots (*Figure 2—figure supplement 1*), which hints at a role of TGN/EE ion and pH regulation in early cell development. *ccc1* plants further exhibit highly increased shoot branching, which gives the knockout plants a bushy appearance (*Henderson et al., 2018*). One cause for this might be defects in auxin transport regulation in shoots. We found alterations in PIN2 exocytosis in *ccc1* roots, suggesting that similar defects could exist in shoot cells (*Figure 5*).

Previous work has revealed a link between endomembrane trafficking, the TGN/EE and salt tolerance in plants. Knockouts of the *Arabidopsis* Rab GTPase ARA6, involved in TGN/EE to PM trafficking, result in salt-sensitive plants. Likewise, knockouts of VACUOLAR PROTEIN SORTING 4 (VPS4), involved in sorting of proteins for vacuolar degradation, result in salt-sensitive plants. Similarly, both *det3* plants and RNAi knockdowns of the TGN/EE V-H$^+$-ATPase display a stronger reduction in primary root growth in response to 50 mM NaCl compared with wildtype plants (*Batelli et al., 2007*; *Krebs et al., 2010*). *nhx5/nhx6* plants exhibit severe reductions in growth when transferred to media containing 100 mM NaCl and germination of *nhx5/nhx6* seeds, with a reduced TGN/EE luminal pH is almost completely abolished on media containing 100 mM NaCl (*Bassil et al., 2011*). CCC1 appears to play a contrasting role to NHX5/NHX6, as here, we found that the germination of *ccc1* seeds was more tolerant to salt and osmotic stress than wildtype (*Figure 4*). The contrasting roles in osmotic tolerance may be explained by the contrasting roles of the proteins in the TGN/EE. *nhx5/nhx6* knockouts have a lower TGN/EE pH and are more resistant to monensin, while *ccc1* knockouts have a higher TGN/EE pH and are hypersensitive to monensin. NHX5 and NHX6 import K$^+$ from the cytosol and therefore knockouts are likely to have a reduced accumulation of K$^+$ in the TGN/EE whereas we find here that CCC1 is likely to export K$^+$ and therefore knockouts likely have an increased accumulation of K$^+$. The elevated TGN/EE luminal pH in *ccc1* under control condition might further prime them for osmotic shock treatments that induce plasmolysis, as luminal pH is as high in *ccc1* under control conditions as it is in wildtype in response to osmotic shock.

CCC1 is important for root hair growth under standard conditions, but *ccc1* knockout root hair growth occurred at the same rate as wildtype at an elevated osmolarity. The greater germination rate of *ccc1* seeds on media with a higher osmolarity may be the result of a combination of factors, including changes in TGN/EE luminal pH and multiple downstream factors. Changes in the cell wall composition of *ccc1* knockouts have been observed (*Han et al., 2020*), so *ccc1* plants might have a weaker cell wall and potentially a weaker seed coat, which might require a smaller water uptake for the necessary seed swelling and seed coat rupture to occur.

Knockouts of *ccc1* require higher concentrations to induce plasmolysis of root epidermal cells. This increased tolerance to plasmolysis could occur if *ccc1* knockouts were able to regulate their internal osmolarity faster than wildtype. It is possible that *ccc1* root epidermal cells could influx external osmolytes faster, due to altered PM localisation of osmolyte transporters, to more rapidly match the external osmolarity. Supporting this, nutrient uptake and ion translocation have been shown to be affected in *ccc1* plants, with altered total shoot K$^+$, Na$^+$, and Cl$^-$ accumulation compared to the wildtype (*Colmenero-Flores et al., 2007*; *Henderson et al., 2015*). Interestingly, *Osccc1.1* roots have a lower cell sap osmolality, but an increase in biomass was reported when plants grown in saline conditions compared to control conditions (*Chen et al., 2016*). The model we propose for *Arabidopsis* CCC1 might therefore be similar in rice, suggesting a conserved function.

Here, we show the TGN/EE pH of epidermal root cells increases in response to both salt and osmotic shock, which has not been reported previously. Modulation of the TGN/EE pH in response to abiotic stress may enable wildtype plants to rapidly regulate protein trafficking and abundance. *det3*, *nhx5/nhx6*, and *ccc1* plants all exhibit TGN/EE pH differences compared to the wildtype of around 0.3 units (*Luo et al., 2015*; *Reguera et al., 2015*; *Dragwidge et al., 2018*; *Dragwidge et al., 2019*), very similar to what we consistently found for *ccc1*. All three genotypes exhibit altered responses to

salt and osmotic treatments, suggesting that a dynamic but tight regulation of TGN/EE luminal pH is an important factor for plant response to these abiotic stresses.

In conclusion, our research here demonstrates that CCC1 has a central function in plant cells and fills a gap in our understanding of how endomembrane luminal pH is regulated through the identification of a novel ion efflux component of the TGN/EE ion transport network.

## Materials and methods

### Plant material and growth conditions

*A. thaliana* plants were all in the Columbia-0 (Col-0) background. Previously described T-DNA insertion lines in AT1G30450, *ccc1-1* (SALK-048175), and *ccc1-2* (SALK-145300) were used in this study (*Colmenero-Flores et al., 2007*). *PIN2::PIN2-GFP, 35 S::VHAa1-RFP, UBQ10::PIP2;1-GFP, 35 S::GFP-LTIB6,* and *35 s::SYP61-pHusion* constructs and plant lines were previously described (*Cutler et al., 2000*; *Xu and Scheres, 2005*; *Dettmer et al., 2006*; *Luo et al., 2015*).

*Arabidopsis* plants were grown on half strength Murashige and Skoog (1/2 MS) media containing 0.1% sucrose, 0.6% phytagel, pH 5.6 with KOH. Plants were sown on plates, incubated at 4°C for at least 2 days and subsequently grown vertically at 21°C and 19°C in 16 hr light and 8 hr dark, respectively; or 8 hr light and 16 hr dark when short day is indicated. Plants were grown for different periods of time, as indicated below and in figure legends. Plants in pots (*Figure 2*) were grown in Irish Peat soil.

### Promoter activity analysis by GUS and Venus fluorescence

GUS staining was done according to *Jefferson et al., 1987*. In summary, plants were submerged in GUS staining solution and stained (plant age and staining times indicated in figure legend). Image of the entire rosette was captured with a Nikon digital camera, flower and inflorescence images with a Nikon SMZ25 stereo microscope. Fluorescence of the nuclear-localised NLS-Venus was imaged in plants ranging from 5 to 8 days to 8 weeks as indicated in figure legend. Excitation light wavelength was 514 nm, emission was detected at 520–560 nm, using either a Nikon A1R or an Olympus FV3000 Confocal Laser-Scanning Microscope; with the following objectives: 20× Plan Apo Lambda and 40× Apo LWD WI Lambda S (Nikon), and 10× UPSLAPO objective (Olympus).

### Root hair length, root hair elongation rate

Light microscopy imaging of root hair length was performed using a Nikon SMZ25 stereo microscope with a 2× objective. For quantification of root hair length, images of roots were taken from above the maturation zone of 6-day plants. Each measurement was of a single root hair. Multiple root hairs were measured per plant. Root hair length was measured using FIJI (*Schindelin et al., 2012*). For time-lapse light microscopy of root hair elongation rate, plants were germinated in 2 ml of media placed in one-well microscopy slides (Thermo Fisher) and grown vertically. Images of root hairs in the maturation zone were taken every 30 s for 6 hr using a Nikon Diaphot 300. Measurements were taken from the beginning of root hair elongation, of root hairs that elongated beyond the initiation phase, until root hair growth ceased. For consistency, elongation rates of root hairs were only measured for root hairs where both initiation and cessation of growth could be observed in the time-lapse. A single root hair was measured per plant. Analysis and creation of videos was performed using FIJI.

### Root morphology imaging

Root morphology images for epidermal cell length and ectopic root hairs were taken at the same Nikon confocal, using 6-day seedlings and root cell wall autofluorescence (excitation 404 nm, emission 425–475 nm). Each measurement for epidermal cell length was of a single cell. Multiple cells were measured per plant. Adjacent trichoblasts were also imaged in root epidermal cells of the maturation zone of *ccc1* plants expressing the trichoblast marker *PRP3::H2B-2xmCherry* (excitation = 561 nm, emission = 570–620 nm; *Marquès-Bueno et al., 2016*).

### GFP-CCC1 cloning and expression

For stable expression of *CCC1* in root hairs, 1402 bp of the trichoblast-specific promoter *EXP7* (*Marquès-Bueno et al., 2016*) was first amplified from Col-0 genomic DNA, using the primers

EXP7pro-HindIII_F (tatacAAGCTTATTACAAAGGGGAAATTTAGGT) and EXP7pro-KpnI_R (cttat GGTACCTCTAGCCTCTTTTTCTTTATTC), following a Phusion PCR protocol (NEB). The PCR product and the binary plasmid pMDC43 were subsequently cut with the restriction enzymes *Hind*III-HF and *Kpn*I-HF to remove the *2 × 35*S promoter, and the digestion reactions were purified using illustra GFX PCR DNA and Gel Band Purification kits. Fragment ligation was performed using T4 DNA Ligase protocol (NEB) at 16°C overnight; two µl of the ligation reaction was transformed into *Escherichia coli* DB3.1 cells and after a sequencing verification, a plasmid was subsequently selected that showed the correct replacement of the *2 × 35*S promoter with the *EXP7* promoter. *CCC1* CDS (with stop codon) was then shuttled into the pMDC43EXP7 using LR clonase II enzyme, which creates N-terminally *GFP*-tagged *CCC1*. Correct plasmids were transformed into *Agrobacterium tumefaciens*, and heterozygous *Arabidopsis* plants (*ccc1*+/-) were floral dipped as the homozygous *ccc1* knockout does not support floral dipping well. Floral dipping was performed according to *Clough and Bent, 1998*, and transformants were selected on 1/2 MS plates with no sucrose, containing hygromycin for selection. Homozygous *ccc1* knockouts expressing *GFP-CCC1* were selected, and phenotyped in the next generation.

## Colocalisation

For colocalisation, 6-day plants expressing both *GFP-CCC1* (excitation 488 nm, emission 500–550 nm) and the TGN/EE marker VHAa1-RFP (excitation 561 nm, emission 570–620 nm) were imaged using Nikon A1R Confocal Laser-Scanning microscope, using a 60× Plan Apo VC WI objective with a numerical aperture of 1.2; pinhole set to of 1.2 AU. Five roots were imaged, with three images of three separate mature epidermal root cells being imaged per plant. Analysis was performed on image stacks with a step size of 0.45 µm. Colocalisation was assessed using the FIJI plugin DiAna (*Gilles et al., 2017*). In brief, segmentation was performed using the iterative segmentation function before the number of objects that overlap is counted. The percentage of overlapping objects is reported as the percentage of colocalisation. Colocalisation with DiAna was supported by obtaining the Pearson's coefficient on the same stacks using JACoP (*Bolte and Cordelières, 2006*).

## GFP-CCC1 imaging

Six-day old plants expressing *GFP-CCC1* (excitation = 488 nm, emission = 500–550 nm) driven by the *EXP7* promoter were imaged on a Nikon A1R Confocal Laser-Scanning microscope. For observation of GFP-CCC1 within BFA bodies, roots were first incubated in 1/2 MS with 4 µM FM4-64 (excitation = 561 nm, emission = 570–620 nm) for 5 min before being washed and incubated in 1/2 MS with 25 µM BFA for 1 hr. Stacks were taken of elongating root hairs. For observing if GFP-CCC1 cycles to the PM, plants were incubated for 30 min in 1/2 MS with or without 50 µM tyrphostinA23. Stacks were taken of mature root epidermal cells. For observing if osmotic shock can induce a change of localisation of GFP-CCC1, plants were incubated in 1/2 MS solution for 1 hr before being transferred to milliQ water, 1/2 MS solution or 1/2 MS solution containing 300 mM sorbitol for 5 min before imaging. Stacks were taken of elongating root hairs.

## High-pressure freezing and TEM

Wildtype and *ccc1* mutant seeds were grown on 1/2 MS plates with 1% sucrose in long-day conditions for 5 days. Roots tips were excised using a sharp razor then cryofixed according to *McFarlane et al., 2008*, with modifications. Cryofixation was performed using a Leica EM-ICE high-pressure freezer, type A carriers (Leica), and 1-hexadecene (Sigma) as a cryoprotectant. Roots were freeze-substituted in a Leica AFS2 at –85°C over 5 days in a mixture of 2% osmium tetroxide (Electron Microscopy Sciences) and 8% 2,2-dimethoxypropane (Sigma) in anhydrous acetone. Samples were warmed to room temperature over 2 days, then slowly infiltrated with Spurr's resin over 5 days (*Spurr, 1969*). Samples were polymerized for 36 hr at 70°C, then sectioned to ~70 nm (silver) on a Leica UC7 microtome using a DiATOME knife. Sections were placed onto copper fine-bar grids (Gilder) coated with 0.3% formvar, then post-stained with 1% aqueous uranyl acetate (Polysciences) for 10 min and triple lead (sodium citrate, lead acetate, lead citrate from BDH, lead nitrate from Fisher; *Sato, 1968*) for 4 min. Samples were viewed using a Phillips CM120 BioTWIN transmission electron microscope with tungsten filament at an accelerating voltage of 120 kV and a Gatan MultiScan 791 CCD camera.

## Germination and collet hairs

Both seed germination and collet hair formation was assayed on the same set of seeds. Seeds were germinated on 1/2 MS media containing the indicated concentration of mannitol, sorbitol, or NaCl. The control seeds, germinated on media with no added mannitol, sorbitol or NaCl, were shared between all three treatments. Plants were stratified at 4°C for 2 days, and then transferred to a growth chamber. Six days after transfer, the percentage of seeds that had germinated were counted with radicle emergence being used as the indicator for germination. On the same day, for all plants that had developed to the point where collet hairs should be visible, the percentage with properly developed collet hairs was determined. Development of collet hairs was scored based on four categories, plants with no visible collet hairs, plants with one or two collet hairs present, plants with a cluster of collet hairs but lacking a complete ring of collet hairs and plants with a complete ring of collet hairs. Plants were grown in batches such that the seeds from each genotype used were harvested from the same batch (i.e. harvested within 6 weeks of each other as *ccc1* plants always take significantly longer to flower), three different batches were used in six different experimental repetitions.

## Root cell sap osmolality

For each sample, seeds (approx. 50 per plate) were placed in a row at the top of a 1/2 MS media plate. After 12 days, roots were cut from the shoots and all roots from the same plate placed together in tubes with 100 µl of water. The sealed tubes were heated to 85°C for 20 min. Tubes were cooled before osmolality was measured with a Friske 210 micro-sample osmometer. Final displayed osmolality was adjusted for the fresh weight of the input material.

## Plasmolysis

Mannitol was used to induce plasmolysis of plants grown for 6 days on 1/2 MS without mannitol. The mannitol concentration at which root cells plasmolysed was determined using epidermal cell without root hairs, in plants expressing the PM marker GFP-LTI6b. Plants were transferred from the growth media to a liquid 1/2 MS solution containing different concentrations of mannitol, as indicated in *Figure 3*, 1 hr before counting. Plasmolysed and non-plasmolysed cells were counted under a Nikon Ni-E widefield microscope. From each root, the plasmolysis state of 40 cells was assessed. Cells which had root hairs were not included. A cell was considered as plasmolysed if the corners of the cell had detached/curved. Only cells past the maturation zone of the plant were included. The estimated concentration at which 50% of cells would plasmolyse for each genotype was determined by fitting a Boltzmann sigmoid curve in GraphPad Prism 9 to obtain the 'V50'.

## Trafficking of PIP2;1-GFP

Internalisation of PIP2;1-GFP in response to osmotic stress was measured in epidermal root cells of the elongation zone of 6-day-old plants. Plants were transferred from 1/2 MS media to liquid 1/2 MS containing 100 mM NaCl or 200 mM mannitol 30 min before imaging, or control 1/2 MS. Imaging was performed using a Nikon A1R confocal laser scanning microscope with a 60× Plan Apo VC WI objective. A ratio of PIN2;1-GFP (excitation = 488 nm, emission = 500–550 nm) at the PM and inside the cell was determined by measuring the signal intensity at the PM and inside the cell after background subtraction with FIJI. All results were normalised to wildtype median signal ratio under control conditions.

## Endocytic trafficking and exocytosis

Endocytic trafficking was assayed in the root tips of 6-day plants using the fluorescent membrane stain FM4-64 (excitation 561 nm, emission 570–620 nm) and the endomembrane trafficking inhibitor BFA. Plants were either incubated in 1/2 MS containing 4 µM FM4-64 and 25 µM BFA in the dark for 10 min before imaging or for 10 min in 1/2 MS with 4 µM FM4-64 and 25 µM BFA in the dark before washing and incubating in 1/2 MS with 25 µM BFA for 50 min before imaging. Images were single representative optical sections, taken of epidermal cells in the root elongation zone. A ratio of cytoplasm:PM signal was measured after background subtraction (using the 'Subtract Background' function) in ImageJ by using the polygon selection tool to measure the mean grey value of the entire interior of the cell and divide this by the PM mean grey value, acquired using the segmented line tool (width 1).

Trafficking of FM4-64 to the tonoplast was measured by incubating plants in 1/2 MS containing 4 µM FM4-64 for 10 min before washing and incubating in 1/2 MS for 3 hr. Images were single representative optical sections taken of epidermal cells in the root elongation zone. A ratio of tonoplast:PM signal was acquired after background subtraction (using the 'Subtract Background' function) using the segmented line tool for both PM and tonoplast measurement.

Exocytosis was assayed in the root tips of 6-day-old plants using PIN2-GFP or LTI6b-GFP in the epidermis. For the 't0' image point, plants were taken directly off growth media and immediately imaged. Otherwise, plants were treated with 25 µM BFA for 60 min, at which point some plants were imaged to obtain the '60' BFA' images. The rest were washed in liquid 1/2 MS and left to recover for 60 min in liquid 1/2 MS media before imaging for the '60' washout' images. Signal internalisation was measured as described for endocytic trafficking. Imaging was done with the 60× objective and Nikon A1R Confocal Laser-Scanning microscope described above.

## TGN/EE swelling with monensin

TGN/EE swelling was induced using the ionophore, monensin (Sigma). Six-day-old plant roots expressing VHAa1-RFP (excitation 561 nm, emission 570–620 nm) were submerged in 1/2 MS solution with or without 2.5 µM monensin for 15 min before imaging. Imaging was performed on the Nikon A1R Confocal Laser-Scanning microscope and 60× objective described above. Images are single representative optical sections, taken of epidermal cells in the root elongation zone. A single measurement was taken per plant, which included multiple cells. Measuring was done using FIJI. TGN/EE were identified using VHAa1-RFP signal and segmented using the automatic threshold algorithm, 'RenyiEntropy'. The average size of the segmented TGN/EE was measured using 'Analyse Particles'.

## TGN/EE and cytosolic pH measurements

The pH of the TGN/EE was measured as described in *Luo et al., 2015*. In brief, SYP61-pHusion (excitation 488 and 561 nm, emission 500–550 and 570–620 nm) was imaged in epidermal cells of the root elongation zone of 6-day-old plants on the Nikon A1R Confocal Laser-Scanning microscope and 60× objective described above, obtaining a single optical section. GFP/RFP intensity ratios were obtained from cells incubated in solutions of known pH to create a calibration curve. TGN/EE pH was measured by obtaining ratios from control plants. The calibration curve was created by measuring cells treated for 15 min with 50 mM MES-BTP (bis-tris-propane) or 50 mM HEPES-BTP with 50 mM ammonium acetate. Seven points between pH 5.0 and 8.0 were measured for the calibration curve. Six measurements were taken per point for calibration and a calibration was performed before every experiment. The curve was fit using a Boltzmann sigmoidal in GraphPad Prism 8.0. A single measurement was taken per plant, which included multiple cells. The GFP/RFP ratio was obtained in FIJI by firstly segmenting for TGN/EE using the automatic 'RenyiEntropy' threshold on the RFP signal before measuring fluorescent intensity.

The cytosol was measured as described above for the measurement of the TGN/EE with some adjustments. The cytosolic pH sensor pHGFP (excitation 405 and 488 nm, emission 500–550 nm; *Moseyko and Feldman, 2001*) was used. Sections of images used for measurement were manually cropped to remove any nuclei. pH measurements were obtained by acquiring a ratio of 488/405 nm signal intensity. Automatic thresholding in FIJI was performed using the 'default' algorithm on the 488 nm channel.

For TGN/EE and cytosolic pH measurements after treatment with osmotic or salt shock, plants were submerged in 1/2 MS containing 400 mM mannitol or 200 mM NaCl for 15 min before imaging.

## Vacuole pH measurements

The pH of the vacuole was measured using BCECF-AM (Invitrogen) as described by *Luo et al., 2015*. In brief, 6-day-old plants were incubated in the dark for 1 hr in a 1/2 MS solution containing 10 µM BCECF and 0.02% pluronic F-127 (Invitrogen). Plants were washed before being imaged at the excitations 404 and 488 nm, both emissions were collected at 500–550 nm. The ratio between 404 and 488 is used to measure pH. A calibration curve was created by measuring cells treated for 15 min with 50 mM MES-BTP or 50 mM HEPES-BTP with 50 mM ammonium acetate. Seven points between pH 5.0 and 8.0 were measured for the calibration curve. Six measurements were taken per point for calibration and a calibration was performed before every experiment. The curve was fit using a Boltzmann

sigmoidal in GraphPad Prism 8.0. A single measurement was taken per plant which included multiple cells. The 404/488 ratio was obtained in ImageJ by firstly segmenting using the automatic 'default' threshold on the 488 nm channel before measuring fluorescent intensity.

### Cell types

Imaging for pH and monensin measurements, as well as for endo- and exocytosis, was carried out in epidermal cells in the root elongation zone; imaging of GFP-CCC1 for colocalisation was carried out in mature epidermis cells, specifically in trichoblasts.

## Acknowledgements

We thank Melanie Krebs and Karin Schumacher for helpful discussions and for providing seeds expressing the TGN/EE and cytosolic pH sensors, and Chuang Wang for the construction of PIP2;1-GFP seeds, and Anthony Gendall for providing *nhx5 × nhx6* seeds. Matthew Tucker for the 3× VenusNLS plasmid, Steve Tyerman for helpful discussions, Philip Brewer for *PIN2::PIN2-GFP* seeds, and Christian Luschnig for advice on genotyping. We thank Renée Phillips and Marie Beillevert for assistance with lab and plant work. We thank Adelaide Microscopy, especially Gwen Mayo and Jane Sibbons, for support with microscopy; and we thank the University of Melbourne Advanced Microscopy Facility where electron microscopy was conducted. We thank the Australian Research Council for funding this work through DE160100804 to SW, FT130100709 and CE140100008 to MG, and DE170100054 to HEM; HEM is also supported in part by funding from the CRC program as the Canada Research Chair in Plant Cell Biology.

## Additional information

### Competing interests

Heather E McFarlane: Reviewing editor, eLife. The other authors declare that no competing interests exist.

### Funding

| Funder | Grant reference number | Author |
| --- | --- | --- |
| Australian Research Council | DE160100804 | Stefanie Wege |
| Australian Research Council | FT130100709 | Matthew Gilliham |
| Australian Research Council | CE140100008 | Matthew Gilliham |
| Australian Research Council | DE170100054 | Heather E McFarlane |
| Canada Research Chairs Program | | Heather E McFarlane |

The funders had no role in study design, data collection and interpretation, or the decision to submit the work for publication.

### Author contributions

Daniel W McKay, Formal analysis, Investigation, Methodology, Visualization, Writing – original draft, Writing – review and editing; Heather E McFarlane, Investigation, Methodology, Visualization, Writing – review and editing; Yue Qu, Apriadi Situmorang, Investigation, Methodology; Matthew Gilliham, Conceptualization, Funding acquisition, Methodology, Resources, Supervision, Validation, Writing – original draft, Writing – review and editing; Stefanie Wege, Conceptualization, Funding acquisition, Investigation, Methodology, Project administration, Resources, Supervision, Validation, Writing – original draft, Writing – review and editing

**Author ORCIDs**
Daniel W McKay http://orcid.org/0000-0001-9743-4961
Heather E McFarlane http://orcid.org/0000-0001-5569-5151
Yue Qu http://orcid.org/0000-0003-0101-1097
Apriadi Situmorang http://orcid.org/0000-0003-1879-3355
Matthew Gilliham http://orcid.org/0000-0003-0666-3078
Stefanie Wege http://orcid.org/0000-0002-7232-5889

**Decision letter and Author response**
Decision letter https://doi.org/10.7554/eLife.70701.sa1
Author response https://doi.org/10.7554/eLife.70701.sa2

## Additional files

**Supplementary files**
• Supplementary file 1. Supplementary information.
(a) Experimental approaches taken to detect subcellular localisation of CCC1 protein. (b) Strategies to isolate a double *ccc1-2 det3* or a triple *ccc1-2 nhx5-2 nhx6-3* mutant

• Transparent reporting form

**Data availability**
Data generated or analysed during this study are included in the manuscript and supporting file; Source Data files have been provided for Figures 2-7. Original images files can be accessed on figshare.

The following datasets were generated:

| Author(s) | Year | Dataset title | Dataset URL | Database and Identifier |
|---|---|---|---|---|
| McKay D, Wege S, Gilliham M | 2021 | Tissue specific expression CCC1 promoter | https://doi.org/10.25909/17256545.v1 | figshare, 10.25909/17256545.v1 |
| McKay D, Gilliham M, Wege S | 2021 | Root hair branching examples | https://doi.org/10.25909/17256515.v1 | figshare, 10.25909/17256515.v1 |
| McKay D, Wege S, Gilliham M | 2021 | Monensin experiment | https://doi.org/10.25909/17256467.v1 | figshare, 10.25909/17256467.v1 |
| McKay D, Wege S, Gilliham M | 2021 | Cell File Organisation | https://doi.org/10.25909/17256428.v1 | figshare, 10.25909/17256428.v1 |

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
