## [Editor Report]

This paper reports compelling evidence that the single isoform of cation chloride cotransporter (CCC1) encoded in *Arabidopsis thaliana* provides a cation and anion efflux mechanism to regulate pH in the trans-Golgi/Early Endosome Network.

---

## [Decision Letter]

**Decision letter after peer review:**

[Editors’ note: the authors submitted for reconsideration following the decision after peer review. What follows is the decision letter after the first round of review.]

Thank you for submitting the paper "Plant Trans-Golgi Network/Early Endosome pH regulation requires Cation Chloride Cotransporter (CCC1)" for consideration by *eLife*. Your article has been reviewed by 2 peer reviewers, and the evaluation has been overseen by a Reviewing Editor and a Senior Editor. The reviewers have opted to remain anonymous.

We are sorry to say that, after consultation with the reviewers, we have decided that this work will not be considered further for publication by *eLife*.

Thank-you for submitting your work to *eLife*, which has now been evaluated and discussed by two peer reviewers and myself. We all enjoyed reading your work and especially liked the pH sensor experiments. While all of us agree that your work is well done overall and will be of interest to a wide audience, we have some important reservations about the major conclusions of the study. We have provided some ideas for how to address these concerns below, but as they will require careful experimentation I'm afraid we cannot accept the manuscript at this time. If you are able to sufficiently address the concerns raised, I would be happy to re-evaluate this work again at a future date.

The major concerns have to do with:

1) The functionality of the GFP-CCC1 construct used, and

2) The data supporting a role for CCC1 in endocytosis.

Regarding the GFP-CCC1 construct:

A major part of your model is that CCC1 resides in the TGN/EE, and it is clear that you tried multiple constructs in order to assess this. However, beyond anecdote, none of these data are provided in the manuscript. As you transformed ccc1 -/+ plants it should be possible to then have assessed dose-dependency, or calculated chi-squared values that show that expression of the various constructs you attempted are lethal. The construct you did use in the paper is expressed under the control of a tissue-specific promoter (pExp7), a clever approach. This construct is able to complement ccc1 defects in root hair growth, although it is not clear if the trichoblast identity defect is complemented. We suggest a more thorough cataloging of complemented and non-complemented ccc1 phenotypes in various tissues. Furthermore, because you were unable to assess complementation of native promoter-driven GFP-CCC1 fusion in ccc1 mutants, it is possible that the expression of an N-terminal construct is dominant-negative. We appreciate that this is difficult to assess, and that you have tried many constructs already, but the localization of CCC1 in the TGN/EE underscores all of your conclusions. One of the reviewers suggests some other tools available to test localization in vegetative tissues. We are also concerned that the N-terminal tag may alter the localization (and/or visualization) of CCC1. Furthermore, as the co-localization between CCC1 and VHA-a1 suggests that CCC1 additionally localizes outside of the TGN/EE, it seems essential to conduct co-localization studies with additional markers.

Regarding endocytosis:

Several concerns were raised regarding endomembrane trafficking experiments. In particular, how defects were quantified and interpreted, and the potential need for more markers. Both reviewers provide ideas for how to address these concerns.

In addition, we also suggest that you add important information regarding the material you used:

1) You write that the higher-order mutants you generated ccc1 det3 and ccc1 nhx5 nxh6 are lethal. Please provide the chi-squared values from analysis of F2 segregants that support this claim, and also provide information about how the crosses were conducted (specific alleles, homozygous/heterozygous lines crossed, etc).

2) It is not clear if the trafficking marker lines you analyzed express similarly in all backgrounds, nor if they are homozygous. Even though crossing should mitigate any positional effects on gene expression, this should be confirmed by westerns or RT-PCR. This is an important control when evaluating differences in marker intensities.

*Reviewer #1:*

In their manuscript "Plant Trans-Golgi Network/Early Endosome pH regulation requires Cation Chloride Cotransporter (CCC1)" the authors sought out to understand the importance of the cation chloride co-transporter CCC1 on plant function and intracellular ion homeostasis. The authors provide new data showing that CCC1 functions at the TGN/EE where it regulates ion homeostasis. Plants lacking CCC1 show a disruption to normal endomembrane trafficking, leading to defects in root hair cell elongation and patterning. Interestingly the authors show that the cell elongation defects can be rescued by supplementing the plants with an external osmolyte such as mannitol. Through the characterisation of CCC1 in *A. thaliana*, this paper shows that cation/anion transporters are essential in maintaining fine control over endosomal pH, in addition to previously characterised endosomal proton/cation transporters such as NHX5, NHX6, and CLCd.

The paper is well written, and the experimental design is generally well thought out. The data mostly supports the authors conclusions, however there are some areas where changes are necessary to improve the clarity and completeness of the experimental work.

1) The co-localisation experiment of CCC1 with VHA-a1 (TGN/EE marker) shows that they highly overlap, however, there are clear regions where the CCC1 and VHA-a1 marker do not co-localise, suggesting CCC1 has a broader localisation pattern which is also alluded to in the text.

It is important to clearly determine which endomembrane compartments CCC1 localises to as this has large implications in interpretation of data regarding where the endomembrane trafficking defects originate from (eg: TGN/EE dysfunction, or other organelles, such as the Golgi and MVB/LE), and for comparisons with other intracellular transporters such as NHX5 and NHX6 (which have broader localisation at the Golgi, TGN/EE, and MVB/LE). A more detailed localisation approach by also assessing the co-localisation of CCC1 with Golgi and MVB/LE markers is necessary.

2) The authors identify defects in cell elongation in ccc1-1 root epidermal cells, as well as defects in the formation of collet hairs. It is not clear whether the defects in collet hair formation is due to defects in cell elongation, or in root hair cell identity as root hair cell identity is disrupted in ccc mutants. Since under control conditions some ccc mutants do not form collet hair cells at all this would suggest that the hair cell identity is also disrupted, rather than just elongation. However, the root hair length quantification experiment does show very clear cell elongation defects in ccc1 mutants. The two phenotypes should be differentiated more clearly in the text.

3) Figure 6 describes experiments designed to assess whether ccc1 mutants have defects in endo- and/or exocytosis. The authors assess endocytosis using an FM4-64 uptake experiment where they conclude that ccc1 mutants have defects in endocytosis. However, the data from the 10-minute time point (which is usually used to measure endocytosis) shows no difference between wild-type and mutant lines. There are clear differences in FM4-64 uptake to the BFA bodies after 60 minutes (Golgi+TGN) which instead suggests ccc1 mutants primarily have defects in post-Golgi trafficking, rather than endocytosis.

The authors should also assess whether secretion/recycling of PIP2;1 and PIN2-GFP is altered by quantifying the signal at the plasma membrane, and potentially by performing FRAP assays of PIP2;1 or PIN2-GFP at the plasma membrane. The authors could also assess whether ccc1 mutants have general defects in secretion by visualisation of sec-RFP in ccc1 mutants. These experiments (in addition to the co-localisation experiments suggested above) would provide much stronger evidence to determine the exact source of trafficking defects.

4) The calibration curves from Figure 7 are missing.

5) The control image of PRP3::H2B in wild type seedlings is missing.

Suggestions for improved or additional experiments, data or analyses,

The authors state that "an essential component required to complete the TGN/EE membrane transport circuit remains unidentified − a pathway for cation and anion efflux", however it is not clear why the cation flux via CCCs is an essential component. Are similar proteins found and necessary in other biological systems (eg: animal or yeast system), and do all plant species contain CCC genes?

Figure 6A. The signal in the ccc mutant after BFA washout seems reduced. Can the authors assess whether PIN2 fluorescence intensity at the plasma membrane is normal, since nhx5 nhx6 mutants have reduced PIN2 plasma membrane localisation. The authors state in the discussion that altered PIN2 trafficking may cause the bushy phenotype in the ccc1 mutants so this should be investigated in more detail.

The monensin experiment with the TGN/EE size quantification is not convincing and needs validation. It is not conclusive whether the TGN/EE size can be accurately quantified with a standard confocal setup, since TGN/EE spots are diffraction limited and the signal is likely very weak after monensin induced swelling. Can the authors provide higher resolution images from these panels and show how the quantification was performed?

The authors exclude outliers in some of their analysis, but it is not justified in the text why this was done. This is likely not appropriate, and outliers should be included in the box plots and used for the analysis.

In Figure 6A It is stated that there are defects in PIN2 exocytosis, however it is not very clear from the images. From the image it seems that the PIN2 may have lower intensity at the PM after washout. The authors should also re-analyse the data to show whether PIN2-GFP plasma membrane levels are similar between WT and ccc1 mutants, or if there is reduced PM intensity as in nhx5 nhx6 mutants. How was the calculation of signal ratio performed? Was the background subtracted from analysis? Assessing the intensity by including only the top 10% of pixels for analysis may provide more reproducible and accurate results.

Figure S1 – please quantify the proportion of the ccc1 roots which have branching/forking. Was this seen in both ccc1 T-DNA lines?

Figure 2C – the order of treatments does not correspond to the order in the figure. Determining whether the localisation changes would be better shown by performing this experiment in the GFP-CCC1 and VHA-a1-mRFP double reporter line.

Recommendations for improving the writing and presentation.

There are many figures where the image quality is too low to determine whether the data supports the conclusions. For example, Figure 5D, 7A. This makes it very difficult to determine whether there are real differences and/or whether the authors have appropriately quantified the data. It is sometimes not possible to visualise differences in between samples because the image is too zoomed out (eg: Figure 7A), or too small (eg: 7B-F). In some cases using a different lookup table (like in Figure 7B-F) would greatly help visualising differences in signal intensity. A large improvement in presentation of the images is necessary.

The discussion/conclusions of this paper are not very clear. From reading the discussion the authors have identified defects in trafficking of PIN2, PIP2;1, and delayed vacuolar trafficking. Comparing these results to nhx5 nhx6 and clc mutants suggests that general TGN/EE pH ion dysfunction causes broad defects in protein trafficking. It is not clear in the discussion why external osmolality rescues the cell elongation defects and the significance of this.

The authors sometimes make statements regarding that one sample /tissue is brighter than another, however I do not think these types of qualitative statements are appropriate, and should be supported by quantitative data or removed.

Figures

Scale bar sizes should not be shown in the image, but rather described in the figure legend, as otherwise it is too small to see.

Figure 4.

The quantification of panel C is in B. The order should be switched of these two panels. (Same as E and F).

The manuscript would flow better by starting with the clear cell elongation defects and rescue in panel G and H. The panel legend may also need to be changed to root cell elongation and cell identity defects.

Figure 5. (D) Cannot see any differences between the WT and mutant. A heatmap LUT may better show differences in image intensity.

Figure 7. (A) This image is way too small to see any differences between the two genotypes. A zoomed in image would be better.

*Reviewer #2:*

In the submitted paper, the authors first show that activity of the CCC1 promoter is ubiquitous. They further analyze the phenotype of the mutant in the root and show a root cell elongation defect in epidermal cells as well as in root hairs. The ccc1 mutants also lack the collet root hairs and show trichoblast-atrichoblast cell fate identity defects in the primary root. The authors perform a set of elegant experiments where they show that, surprisingly, the ccc1 plants are resistant to hyperosmotic environment. The ccc1 cells show delayed plasmolysis, ccc1 seeds show better germination, and ccc1 root hair elongation is recovered in hyper-osmotic media. Interestingly, the absence of collet root hairs was also recovered in hyper-osmotic environment, even though it is not clear whether this was caused by 'reparation' of collet hair elongation or collet hair cell fate specification. The phenotypic analysis is carefully performed and the results are unexpected and intriguing.

The authors further show that in root trichoblasts, GFP-CCC1 localizes to the TGN/EE compartment, and that in this tissue, the fusion protein recovers the root hair elongation of the mutant. Further, the authors focus on the subcellular phenotypes of the endomembrane system performance in the ccc1 mutant background. It is shown that PIP2 aquaporin internalized less in the ccc1 than in control, which hints to that endocytosis is reduced in the ccc1 cells. An alternative explanation however could be that the mutant is more osmotically tolerant also on the subcellular level. To test the endomembrane trafficking rate, PIN2 aggregation and recovery from BFA bodies is performed, as well as quantifications of FM4-64 uptake, and the authors conclude that the mutant has generic endomembrane trafficking defects.

The authors hypothesize that the endomembrane defects might stem from a disturbance in TGN/EE luminal pH caused by an ion imbalance in the ccc1 cells. Therefore, they measured the luminal pH of TGN/EE using a genetically encoded phluorin and demonstrate a more alkaline pH values in the ccc1 mutants. Finally, the authors show that during hyperosmotic stress, the TGN/EE pH rises in the control plants, suggesting that this pH rise is functionally connected to the stress response. The second part of the manuscript that focuses on subcellular phenotypes uses advanced live-cell imaging tool and successfully measures pH in minute volumes of TGN/EE compartments. In addition, the specificity of the phenotype is demonstrated by careful analysis of vacuolar and cytoplasmic pH. These well performed experiments indeed point to the function of CCC1 in ion control in TGN/EE.

Weaknesses of the manuscript:

The functionality of the GFP-CCC1 fusion is questionable as it was impossible to obtain transgenic lines that would express GFP-CCC1 under the control of the native promoter, not allowing full complementation of the ccc1 phenotype. This hints to a possible dominant-negative effect of this particular protein fusion. The authors therefore express GFP-CCC1 using a trichoblast-specific promoter and show that the root hair elongation phenotype is complemented, demonstrating some functionality of this construct. Moreover, the root hair length data in the ccc1-1 mutants shown in figure 2D and 3C differ, which to some extent weakens the important conclusion that the GFP-CCC1 is functional at least in this cell type. Functionality of this construct is a crucial aspect for the manuscript. The possible dominant-negative effect of the construct weakens the conclusion about the subcellular protein localization, which in turn weakens the main conclusion of the paper – that CCC1 by regulating ion fluxes in the TGN/EE allows proper endomembrane functionality. The subcellular localization of CCC1 should be demonstrated without any doubts, as it was previously localized to the PM and endomembranes in pollen tubes (Domingos 2019). If CCC1 localized at the PM, alternative explanations of the phenotypes of the nature of the mutant phenotype that would include regulation ion fluxes across the PM would appear more probable than the TGN/EE hypothesis.

Quantification of endomembrane trafficking represents another important argument in the proposed hypothesis. The section that demonstrates the reduction in exo- and endocytosis is however not utterly convincing. It has been shown that a major contribution to the BFA body PIN2 pool originates from de-novo synthesis of the protein (Jasik et al., PMID:27506239). In the figure 6A, it is apparent that the BFA washout leads to disappearance of BFA bodies in the ccc1 mutant, but the level of PM fluorescence was decreased, leading to an apparent 'minimal recovery' of the cytoplasm:PM ratio. In case of endocytosis, the experiment combining FM4-64 uptake with BFA is hard to interpret as endocytosis visualization, because TGN/EE aggregation might be disturbed in the ccc1, as the authors suggest. A more detailed endomembrane trafficking then simple cytoplasm/PM ratios of signal could be performed to address what is happening with trafficking in this interesting mutant.

Suggestions:

It might be helpful to analyze the subcellular localization of CCC1-GFP in the vegetative tissues in the Lat52-driven constructs (Domingos 2019). Lat52 can be leaky in the vegetative tissues and has been used to drive mild expression of proteins such as T-PLATE (PMID: 24529374). This could help strengthen the claim that CCC1 is localized to TGN. The Lat52 driven construct in the Domingos paper was constructed from genomic sequence, contains introns, which is another difference from the CDS construct used in the submitted manuscript.

What is the topology of the CCC1 in the membrane – where are the GFP fusions localized? Could the problem of the non-shining internal GFP fusions be the low luminal (or apoplastic in the case of PM localization of the protein) pH of the compartment that would quench the GFP fluorescence? This could be tested by rising the pH of the endomembranes of the transgenic lines, or by using a pH-insensitive fluorescent protein.

Comments:

Student t-test for comparing multiple samples is repeatedly used. Appropriate statistical tests for comparing multiple samples should be used instead, and also the data distribution should be tested for normality in cases when parametric tests are used.

The authors explain the short root phenotype by a reduction of cell length. However, from the data provided, the cell length seems to be far less reduced than the overall root growth.

Trichoblast identities: Is the trichoblast identity problem complemented in the trichoblast-CCC1-expressing lines?

Quantification of endocytosis and endomembrane trafficking is not very clear to me: how can the cytoplasm/PM ratio of fluorescence give the value of 1, when there is almost no fluorescence in the cytoplasm in the 10 min of FM4-64 uptake or in the T0 in case of the PIN2 GFP fluorescence? Also, not always same areas of roots are shown for treatments and mutants (authors state that data was quantified in elongation zone, but images show rather meristem/transition zone). Were the quantifications of the fluorescence signals done blindly? Do not take this as an offense, but from my experience, in case of data, where regions of interest are drawn by hand, experiments should be done so that the researcher imaging the samples at the microscope does not know which genotype they are imaging, as the FM signal in different root zones and z sections is very variable. I would suggest a simple pulse-chase FM4-64 uptake experiment without including BFA to reveal the dynamics of FM uptake in the mutant to provide more convincing data showing the endocytotic defect.

[Editors’ note: further revisions were suggested prior to acceptance, as described below.]

Thank you for choosing to send your work entitled "Plant Trans-Golgi Network/Early Endosome pH regulation requires Cation Chloride Cotransporter (CCC1)" for consideration at *eLife*. We appreciate your constructive and thorough letter of appeal, which has been discussed between both reviewers and the Reviewing Editor. We are prepared to consider a revised submission, however please note that there are no guarantees of acceptance.

We remain enthusiastic about your work and welcome you to submit a revised manuscript along the lines of what you outlined in your rebuttal. Our original decision was based on the expected time needed to perform more conclusive experiments, but you have convinced us that many of our concerns can be addressed with thorough discussion, more precise language, and a few experiments that will only take 1-2 months.

Regarding our main concerns:

Functionality: As you could not obtain transformants in most cases, we still think this unfortunately means that tagged CCC1 is not fully functional. However, we also agree that your focus on the complementation of a cellular phenotype is sound and justified. We would ask that you do provide a more thorough cataloging of complemented and non-complemented phenotypes (you mentioned in your letter that these data are available, at least with respect to trichoblast identity), and further discussion on this construct in particular, and in comparison to others, as outlined in your rebuttal. In addition, a more detailed discussion of the issue and a more detailed overview (in a Supplemental table, as you suggest) of what has been tried to obtain the functional transformants would demonstrate that everything possible was tried. In addition, a more pointed discussion about the localization of CCC1 as found in proteomics studies should be included.

Endocytic trafficking: We agree that a more careful wording will help clarify distractions and focus on the function of CCC1 in the TGN/EE and its role in ion regulation. However, the data presented to determine defects in endocytic trafficking may still need some re-analysis: for example, quantification of PM and BFA signals. To help clarify the defect in trafficking, it would be useful to do a FM pulse chase experiment to complement the approach you show in Figure 6E-F. One reviewer suggests that the simplest way to improve this section would be to remove Figure 6C 10 min time point (BFA + FM) and replace this with an FM pulse-chase experiment (FM treatment for 10-15 min without BFA) to test for defects in endocytosis/trafficking more clearly. Quantification of the plasma membrane signal in Figure 6A alone would also help with the interpretations of the BFA washout experiment (i.e., if the PIN2-GFP signal is significantly reduced in the mutant this may be an interesting result as well).

Outliers: Thank-you for bringing this to our attention. We apologize for misinterpreting the wording in the legend.

---

## [Author Response]

[Editors’ note: The authors appealed the original decision. What follows is the authors’ response to the first round of review.]

The major concerns have to do with:1) The functionality of the GFP-CCC1 construct used, and2) The data supporting a role for CCC1 in endocytosis.Regarding the GFP-CCC1 construct:A major part of your model is that CCC1 resides in the TGN/EE, and it is clear that you tried multiple constructs in order to assess this. However, beyond anecdote, none of these data are provided in the manuscript. As you transformed ccc1 -/+ plants it should be possible to then have assessed dose-dependency, or calculated chi-squared values that show that expression of the various constructs you attempted are lethal.

The model we present relies on CCC1 being localised in the TGN/EE. We show complementation of a cellular phenotype (root hair elongation) and consequences of loss of CCC1 (disturbed pH and osmolyte regulation in the TGN/EE) that further reinforces that the protein has a functional role in the TGN/EE. The observed defects in TGN/EE function of *ccc1* plants requires the fewest assumptions, if CCC1 is present upon and functions in the TGN/EE; this includes a role in regulation of luminal pH and osmolyte control, as well as the trafficking defects. Not only is the model supported by our own data but also multiple proteomic studies that show that CCC1 is highly abundant in TGN/EE. Proteomics from Sadowski et al. (2008), Drakakaki et al. (2012), Groen et al. (2014), Nikolovski et al. (2014) and Parsons et al. (2019) demonstrate that CCC1 is localised in the TGN/EE, and was not found on other membranes.

In regard to assessing dose-dependency and lethality via chi-squared, this would ordinarily be an elegant way to characterise a sub-lethal construct. However, it is not possible here, as no transformants could be obtained. This includes in WT, *ccc1* +/- and *ccc1* knockout backgrounds. In contrast, the non-tagged CCC1 could be expressed in both wildtype and *ccc1* knockout plants using the *CCC1* CDS with either the native or *35S* promoter; such expression in the *ccc1* background results in phenotypic complementation. To better explain the multiple attempts we took and their outcomes, we will include a supplementary table with details of all constructs. This will include N- and C-terminal tagging, internal tagging, GFP, mCherry and FLAG tags, different linkers, and a rabbit derived antibody against native CCC1. We think this will better support our statement in the manuscript text.

Finally, we feel it is important to distinguish between the cellular function of CCC1 and its role at the plant level. In this manuscript, we have focused on its cellular function, in particular on the role of the ion transporter in the regulating luminal conditions in the subcellular organelle in which it was found to be most abundant. Therefore, we have documented subcellular localisation using a construct that complements cellular elongation defects caused by CCC1 loss. We agree that a construct that rescues all plant phenotypes of *ccc1* mutants would be important to support claims linking these other plant phenotypic alterations to loss of CCC1, but the focus of our work is on the cellular role of CCC1.

The construct you did use in the paper is expressed under the control of a tissue-specific promoter (pExp7), a clever approach. This construct is able to complement ccc1 defects in root hair growth, although it is not clear if the trichoblast identity defect is complemented.

Thank you for bringing this to our attention. Expression of *GFP-CCC1* by the *EXP7* promoter does not rescue the trichoblast patterning defects in *ccc1* knockouts. The trichoblast result is expected as cell fate is decided earlier than the activation of *EXP7*. We had stated this in a previous draft version of the manuscript; however, this was accidentally lost from the submitted version. We will add this statement back and additionally provide supporting images in the supplementary file, in which the altered trichoblast patterning can be seen in *ccc1* plants expressing *GFP-CCC1* under the *EXP7* promoter.

We suggest a more thorough cataloging of complemented and non-complemented ccc1 phenotypes in various tissues.

Thank you for this suggestion. *EXP7* drives expression in a selection of root epidermal cells (trichoblasts, post-elongation zone). As such, many phenotypic aspects, such as shoot appearance and biomass, root length and trichoblast patterning are not altered by *EXP7* driven *GFP-CCC1* expression. The plants expressing this construct, when grown in pots, resemble *ccc1* KO plants. We are happy to include root length measurements, cell length measurements and shoot biomass measurements in the manuscript to clarify this. These experiments should be relatively easy to include within the scope of revisions.

Furthermore, because you were unable to assess complementation of native promoter-driven GFP-CCC1 fusion in ccc1 mutants, it is possible that the expression of an N-terminal construct is dominant-negative. We appreciate that this is difficult to assess, and that you have tried many constructs already, but the localization of CCC1 in the TGN/EE underscores all of your conclusions. One of the reviewers suggests some other tools available to test localization in vegetative tissues. We are also concerned that the N-terminal tag may alter the localization (and/or visualization) of CCC1.

The constitutive or native promoter driven expression of all tagged constructs is dominant negative. The tools suggested by the reviewer have either already been tried and were unsuccessful, or these avenues have been entirely exhausted, please see detailed comment further below. With reference to the use of LAT52 promoter, we have contacted the Feijo lab, who regularly uses the construct for pollen studies and they have never observed expression in vegetative tissue. Further, the seeds are no longer available following his lab move to the US (we can share the email exchange if the editor requests). Regardless, we kindly suggest that the use of a tomato derived pollen specific promoter to drive weak but uncontrolled expression of the construct in unspecified and putatively varying vegetative tissues and cell types of Arabidopsis is rather convoluted and would not result in an interpretable result fit for publication, particularly regarding the functionality of this CCC1 construct.

In contrast, the construct we use and for which we can provide data, can restore the cellular function of CCC1 when localised in TGN/EE, in a defined cell type in which CCC1 is ordinarily expressed, and loss of the ion transporter leads to documented defects. No functionality or complementation has been shown for the LAT52 driven CCC1 construct in pollen.

Furthermore, as the co-localization between CCC1 and VHA-a1 suggests that CCC1 additionally localizes outside of the TGN/EE, it seems essential to conduct co-localization studies with additional markers.

We agree that the observation that proteins CCC1 and VHA-a1 show excellent but not 100% overlap in localisation is an interesting observation; this is consistent with what has been observed for the other ion transporters proposed to be within the TGN/EE. Studies by other groups have explored TGN/EE heterogeneity and proteins considered TGN/EE resident often do not exhibit complete overlap (See Groen et al. 2014 Journal of Proteome Research, Uemura et al. 2014 Plant Cell Phys., Uemura et al. 2019 Plant Phys., and Heinze et al. 2020 PNAS). Of note, VHA-a1 and SYP61 are both considered markers of the TGN/EE but exhibit a rather moderate degree of co-localisation yet, pHusion-SYP61 is the standard for use in measuring the impacts of VHA-a1 on TGN/EE pH (Groen et al. 2014 Journal of Proteome Research, Luo et al. 2015 Nature Plants). The co-localisation of VHA-a1-RFP and GFP-CCC1 we show is actually extraordinarily high for this organelle (compare SYP43/VHAa1 from Shimizu et al. 2021 Nature Plants, which has only a correlation coefficient of ~0.7 compared to 0.86 for CCC1/VHAa1 in our work). The co-localisation data we show of VHA-a1-RFP with GFP-CCC1 therefore strongly supports our conclusions that CCC1 functions in the TGN/EE.

Regarding endocytosis:Several concerns were raised regarding endomembrane trafficking experiments. In particular, how defects were quantified and interpreted, and the potential need for more markers. Both reviewers provide ideas for how to address these concerns.

We have addressed these in detail below. We noticed that a change in wording will greatly help clarify what we are investigating; we measured how lack of CCC1 impacts endocytotic trafficking, and not specifically the initial endocytosis events at the PM. Similar to what was shown for the proton and other ion transport proteins in the model, we show that lack of ion export in the TGN/EE impacts endocytotic trafficking to the TGN and from the TGN to the vacuole as a result of a disturbed TGN/EE function. Wording will be corrected in the new manuscript version, thank you for bringing this to our attention.

In addition, we also suggest that you add important information regarding the material you used:1) You write that the higher-order mutants you generated ccc1 det3 and ccc1 nhx5 nxh6 are lethal. Please provide the chi-squared values from analysis of F2 segregants that support this claim, and also provide information about how the crosses were conducted (specific alleles, homozygous/heterozygous lines crossed, etc).

We can include a supplementary table detailing the specific alleles and crossing strategies. In brief:

Genetic material used: Homozygous *ccc1-2* allele (Colmenero-Flores et al. 2007, Plant Physiology), homozygous *det3* (Schumacher et al. 1999, Genes and Development), homozygous double mutants *nhx5-2 nhx6-3* (Ashnest et al. 2015, Plant and Cell Physiology). There was no problem obtaining F1 seeds, which were wildtype looking. All three genetic backgrounds are clearly distinct from WT, but WT looking heterozygous plants were expected in F1 as all mutations are recessive. Seeds from F1 plants (=F2 generation) were sown and genotyped, all alleles could be detected in the F2 generation, confirming that all mutant alleles had been successfully passed to the next generation through crossing. However, no double or triple mutants were found in the progeny. We then followed two different strategies, which were dependent on the chromosomal placement of the genes in Arabidopsis.

For *det3 x ccc1-2*, we isolated F2 plants that were homozygous for one gene and heterozygous for the other, for example, *det3* mutants and *ccc1*-/+ and vice versa. This was done because we assumed that any double mutants might be very sick plants and expected a low frequency of these plants. Seeds in which only one of the two mutant alleles was segregating seemed like a more promising approach. We genotyped the progeny of those plants (F3). Yet again, we were not able to recover a double knockout from these plants. Therefore, in the case of *det3 x ccc1*, a chi squared analysis does not enrich the manuscript as the number of double knockouts is zero. We can provide data about the *det3 ccc1 +/+* and *det3 ccc1 +/-* plants obtained from *det3 ccc1 +/-* mother plants, and about the *ccc1 det3* +/+ and *ccc1 det3* +/- plants obtained from a *ccc1 det3* +/- mother plant, if this is still required after our clarification here.

For *nhx5* and *nhx6*, we kindly received the double knockout from Tony Gendall (La Trobe University, Melbourne). These two genes are close to each other on chromosome 1 (AT1G54370 and AT1G79610) and rarely segregate during crosses. In addition, *CCC1* (AT1G30450) is also on the same chromosome arm. Segregation of the *ccc1-2* allele and the *nhx5 nhx6* is therefore not normal and assessment with chi-squared values would be inappropriate, due to genetic linkage. We genotyped many F2 plants and were only able to obtain one triple mutant. This plant did not produce any seed. The genetic linkage of the genes also prevented us from working with seeds from heterozygous plants, as the frequency of triple mutant progeny is too low. As such, as we explained in our letter to the editor prior to submission, analysis and experimentation on *ccc1*/*nhx5/nhx6* triple mutants is not feasible. However, we do have images of the one nonviable triple mutant plant, including images of the empty siliques from that one plant, which we can provide, supporting the statements made in the manuscript.

We think that these explanations are sufficient to support our statement that double and triple knockouts are not viable, and that it is not possible provide the data suggested. We are happy to provide a version of this text in supplementary information if the editors think that this would be instructive.

2) It is not clear if the trafficking marker lines you analyzed express similarly in all backgrounds, nor if they are homozygous. Even though crossing should mitigate any positional effects on gene expression, this should be confirmed by westerns or RT-PCR. This is an important control when evaluating differences in marker intensities.

We agree here that it would be useful to perform qPCR if we were measuring marker intensities, to determine if changes in intensity were coming from changes in protein stability or altered expression. However, in our manuscript we do not present any results where we compare marker intensities, instead we use ratiometric measures, comparing the relative differences in signal distribution. The trafficking assays we perform rely on the use of ratios (Internal/PM) instead of absolute signal intensity, as this controls for changes in signal intensity. Such assays have previously been used in the literature when measuring trafficking of markers with different signal intensities across genotypes, including the constructs and markers we use in our work (Luo et al. 2015, Nature Plants; Dragwidge et al. 2019, Journal of Cell Science). To help clarify this, we can further provide evidence in a supplementary figure, which demonstrates that changes in absolute signal intensity does not impact the ratio. We calculated the correlation coefficient of absolute signal intensity vs ratio and no correlation was found demonstrating that there was no connection between the quantity of the total signal in cells and the distribution within.

The misunderstanding of marker intensities rather than marker distribution may be derived from a sentence referring to the difference in brightness of BFA bodies. This sentence was provided to highlight a possible observed different BFA body “anatomy” (compact and therefore brighter BFA bodies in WT as opposed to more diffuse BFA bodies in *ccc1*). This was not intended as a comparison of marker intensities between WT and *ccc1* but was provided to describe the putative observed difference in BFA body formation that can be seen in the images. We can remove the sentence or move it into the discussion, particularly as we are not aiming to measure differences in BFA body formation. Rather, we used the drug BFA to investigate trafficking defects and hence, mention of BFA body shape and brightness was intended to aid readers in understanding the example images.

Reviewer #1:In their manuscript "Plant Trans-Golgi Network/Early Endosome pH regulation requires Cation Chloride Cotransporter (CCC1)" the authors sought out to understand the importance of the cation chloride co-transporter CCC1 on plant function and intracellular ion homeostasis. The authors provide new data showing that CCC1 functions at the TGN/EE where it regulates ion homeostasis. Plants lacking CCC1 show a disruption to normal endomembrane trafficking, leading to defects in root hair cell elongation and patterning. Interestingly the authors show that the cell elongation defects can be rescued by supplementing the plants with an external osmolyte such as mannitol. Through the characterisation of CCC1 in *A. thaliana*, this paper shows that cation/anion transporters are essential in maintaining fine control over endosomal pH, in addition to previously characterised endosomal proton/cation transporters such as NHX5, NHX6, and CLCd.The paper is well written, and the experimental design is generally well thought out. The data mostly supports the authors conclusions, however there are some areas where changes are necessary to improve the clarity and completeness of the experimental work.1) The co-localisation experiment of CCC1 with VHA-a1 (TGN/EE marker) shows that they highly overlap, however, there are clear regions where the CCC1 and VHA-a1 marker do not co-localise, suggesting CCC1 has a broader localisation pattern which is also alluded to in the text.It is important to clearly determine which endomembrane compartments CCC1 localises to as this has large implications in interpretation of data regarding where the endomembrane trafficking defects originate from (eg: TGN/EE dysfunction, or other organelles, such as the Golgi and MVB/LE), and for comparisons with other intracellular transporters such as NHX5 and NHX6 (which have broader localisation at the Golgi, TGN/EE, and MVB/LE). A more detailed localisation approach by also assessing the co-localisation of CCC1 with Golgi and MVB/LE markers is necessary.

Our data shows that the co-localisation of VHA-a1-RFP and GFP-CCC1 is extraordinarily high at 0.86 for CCC1/VHAa1, compared, for instance, with SYP43/VHAa1 from Shimizu et al. 2021 Nature Plants, which has a correlation coefficient of ~0.7.

Proteomic studies (e.g. Groen et al. 2014 J Proteomic Research), have shown that CCC1 is a high-confidence TGN/EE resident protein, co-localised with VHA-a1 and SYP61. We have included this information in the introduction and results (L86-90 and L182-186). We agree that further co-localisation studies will be useful in the future; at this time point, we focused on the role of CCC1 in the TGN/EE.

2) The authors identify defects in cell elongation in ccc1-1 root epidermal cells, as well as defects in the formation of collet hairs. It is not clear whether the defects in collet hair formation is due to defects in cell elongation, or in root hair cell identity as root hair cell identity is disrupted in ccc mutants. Since under control conditions some ccc mutants do not form collet hair cells at all this would suggest that the hair cell identity is also disrupted, rather than just elongation. However, the root hair length quantification experiment does show very clear cell elongation defects in ccc1 mutants. The two phenotypes should be differentiated more clearly in the text.

Thank you, we have amended the manuscript and now better differentiate between the collet hair phenotype and the root hair phenotype.

We have now included evidence that collet hair elongation, and not cell identity, is disrupted in *ccc1* collet hairs (Figure 4 – supplementary figure 1D). In *ccc1* plants, collet hairs are initiated to some degree but do not elongate, while under increased external osmolarity, collet hairs elongate similar to what was observed in the wildtype.

3) Figure 6 describes experiments designed to assess whether ccc1 mutants have defects in endo- and/or exocytosis. The authors assess endocytosis using an FM4-64 uptake experiment where they conclude that ccc1 mutants have defects in endocytosis. However, the data from the 10-minute time point (which is usually used to measure endocytosis) shows no difference between wild-type and mutant lines. There are clear differences in FM4-64 uptake to the BFA bodies after 60 minutes (Golgi+TGN) which instead suggests ccc1 mutants primarily have defects in post-Golgi trafficking, rather than endocytosis.

We agree with the reviewer’s comment here and we think there was a misunderstanding due to the wording we used. Yes, a 10 minute time point would measure immediate endocytosis and what we quantify at the 60 min time point is endocytic trafficking. We had used the terminology “endocytosis” throughout the manuscript, however, in the wake of these comments we realise that this terminology was not sufficiently precise. As the reviewer correctly points out, what we measured and what we are interested in is “endocytic trafficking”, a process previously shown to be disrupted in mutants with altered TGN/EE pH. We have improved the wording of the manuscript to better reflect this and more strictly adhered to the exact use of endocytic trafficking and endocytosis

The authors should also assess whether secretion/recycling of PIP2;1 and PIN2-GFP is altered by quantifying the signal at the plasma membrane, and potentially by performing FRAP assays of PIP2;1 or PIN2-GFP at the plasma membrane.

We appreciate the reviewers’ interest in this subject; however, the trafficking results are included to support the assertion that CCC1 has a role in TGN/EE pH and ion regulation. Detailed trafficking assays are therefore not a key or central theme in the manuscript and as such, we think that further focusing on the trafficking aspect would distract readers from the primary take home message of the work. Nevertheless, quantification of PIN2-GFP signal at the PM is now included in Figure 6 – supplementary figure 1A, as requested.

The authors could also assess whether ccc1 mutants have general defects in secretion by visualisation of sec-RFP in ccc1 mutants. These experiments (in addition to the co-localisation experiments suggested above) would provide much stronger evidence to determine the exact source of trafficking defects.

We agree that sec-RFP would be another means to assess general secretion defects on-top of what we have already provided. However, we believe that further characterisation of trafficking defects in *ccc1* with sec-RFP will not aid in further determining the exact source of trafficking defects beyond what is already provided in the manuscript. We suggest that the trafficking defects are caused by changes to TGN/EE pH regulation and the mechanism by which pH impacts trafficking is not yet fully understood. To that end, we used FM4-64 and PIN2 to assay trafficking as these are markers used previously to assay trafficking of *det3* and *nhx5/nhx6* mutants. The assessment of CCC1’s role in TGN/EE pH and ion regulation is the central goal of this work.

4) The calibration curves from Figure 7 are missing.

Calibration curves have now been included (Figure 7 – supplementary figure 2).

5) The control image of PRP3::H2B in wild type seedlings is missing.

Calibration curves have now been included (Figure 7 – supplementary figure 2).

Suggestions for improved or additional experiments, data or analyses,The authors state that "an essential component required to complete the TGN/EE membrane transport circuit remains unidentified − a pathway for cation and anion efflux", however it is not clear why the cation flux via CCCs is an essential component. Are similar proteins found and necessary in other biological systems (eg: animal or yeast system), and do all plant species contain CCC genes?

We have re-written the manuscript introduction and discussion to better highlight the importance of CCC proteins in plants and other organisms, such as humans and yeast, and the differences in function between organisms, as well as further highlighting the need of both ion influx and efflux pathways in the endomembrane system.

The monensin experiment with the TGN/EE size quantification is not convincing and needs validation. It is not conclusive whether the TGN/EE size can be accurately quantified with a standard confocal setup, since TGN/EE spots are diffraction limited and the signal is likely very weak after monensin induced swelling. Can the authors provide higher resolution images from these panels and show how the quantification was performed?

We appreciate the concerns raised by the reviewer. Due to concerns of bias, quantification was automated as much as was feasible including the use of auto-thresholding and object identification (Material and Methods L767). The original images (.nd2 files) are publicly available and deposited at https://doi.org/10.25909/17256467 to allow for independent analysis to further alleviate concerns. The consequence of monensin treatment is quite clearly captured in the images, including the difference in response of the genotypes. We have also provided the images with a higher resolution in Figure 7A.

The authors exclude outliers in some of their analysis, but it is not justified in the text why this was done. This is likely not appropriate, and outliers should be included in the box plots and used for the analysis.

We did not remove outliers from any of our analysis. The reviewer’s assumption of outlier exclusion was likely a result of a misunderstanding of the figure legends. We have amended the figure legends to avoid that other readers might misunderstand the representation of our data, too. No data points were excluded from any of our analyses, and if practical (i.e. under 100 data points), individual data points are indicated.

Figure 6A. The signal in the ccc mutant after BFA washout seems reduced. Can the authors assess whether PIN2 fluorescence intensity at the plasma membrane is normal, since nhx5 nhx6 mutants have reduced PIN2 plasma membrane localisation. The authors state in the discussion that altered PIN2 trafficking may cause the bushy phenotype in the ccc1 mutants so this should be investigated in more detail.In Figure 6A It is stated that there are defects in PIN2 exocytosis, however it is not very clear from the images. From the image it seems that the PIN2 may have lower intensity at the PM after washout. The authors should also re-analyse the data to show whether PIN2-GFP plasma membrane levels are similar between WT and ccc1 mutants, or if there is reduced PM intensity as in nhx5 nhx6 mutants. How was the calculation of signal ratio performed? Was the background subtracted from analysis? Assessing the intensity by including only the top 10% of pixels for analysis may provide more reproducible and accurate results.

We have performed quantification of PIN2-GFP signal at the plasma membrane and found that there is a difference between the wildtype and *ccc1* (Figure 6 – supplementary figure 1A); additionally, we have added a graph showing that there is no correlation between overall signal intensity and the signal ratio (Figure 4 – supplementary figure 1B). This means that the use of this assay, through use of the ratio, remains valid.

Yes, background subtraction was used for analysis. This is now mentioned in the Material and Methods (L731, L742 and L748). In regards to using only 10% of pixels for analysis, this can be useful in many cases, however, this would distort the data due to the noted difference in BFA body formation between genotypes and so should not be used in this circumstance.

Figure S1 – please quantify the proportion of the ccc1 roots which have branching/forking. Was this seen in both ccc1 T-DNA lines?

Yes, this was seen in both T-DNA lines. We understand the interest of the reviewer in the root hair branching. These occurrences are so rare that they are extremely difficult to quantify though. The observation was provided due to repeated questioning and interest about root hair branching in *ccc1* mutants we received during seminars and conference talks. This is the reason why we provided the images in the supplementary and not as a main finding in the manuscript.

We have now included a sentence to highlight that malformed root hairs were an extremely low occurrence (L171).

Figure 2C – the order of treatments does not correspond to the order in the figure. Determining whether the localisation changes would be better shown by performing this experiment in the GFP-CCC1 and VHA-a1-mRFP double reporter line.

We are unsure what the reviewer refers to here. The figure referred to has no treatments and rather the mentioned figure contains just Wt/*ccc1*. Figure S2 (now Figure 3 —figure supplement 1) does show some localisation data with treatments. It shows that various treatments do not result in GFP-CCC1 signal at the PM. We request to leave the order of these panels as they are, as we feel they are appropriate.

Recommendations for improving the writing and presentation.There are many figures where the image quality is too low to determine whether the data supports the conclusions. For example, Figure 5D, 7A. This makes it very difficult to determine whether there are real differences and/or whether the authors have appropriately quantified the data. It is sometimes not possible to visualise differences in between samples because the image is too zoomed out (eg: Figure 7A), or too small (eg: 7B-F). In some cases using a different lookup table (like in Figure 7B-F) would greatly help visualising differences in signal intensity. A large improvement in presentation of the images is necessary.

We have changed the images in Figure 5D, Figure 6A, Figure 6C and Figure 6E, fluorescent signals are now shown using the multicolour LUT “fire” to aid in distinguishing features and differences between genotypes/treatments.

Images in Figure 7B-F are highly processed images. We provided them to allow readers to visualise differences in signal ratios (viewed as changes in colour) between genotypes/treatments. As such, the visualisation of structures and features are not vital for these images, the data presented in the accompanying graphs is the key result here.

We have further improved the resolution of the images we uploaded and are happy to supply all our original image data files in an online folder to overcome issues with reduced resolution.

The discussion/conclusions of this paper are not very clear. From reading the discussion the authors have identified defects in trafficking of PIN2, PIP2;1, and delayed vacuolar trafficking. Comparing these results to nhx5 nhx6 and clc mutants suggests that general TGN/EE pH ion dysfunction causes broad defects in protein trafficking. It is not clear in the discussion why external osmolality rescues the cell elongation defects and the significance of this.

We compare the trafficking defects seen in *ccc1* to those in *det3* and *nhx5/nhx6* to support our primary result: CCC1 is important for TGN/EE ion and pH regulation. The main conclusion of work is not that general TGN/EE ion dysfunction causes broad defects; we rather use this known consequence of TGN/EE dysfunction to give further evidence that CCC1 functions in the TGN/EE.

In addition, we show that osmotic stress impacts TGN/EE pH and that *ccc1* mutants have an altered response to osmotic and salt stress like *det3* and *nhx5/nhx6* plants, further supporting this proposed role of CCC1.

The reviewer is right that very little is currently known about the connection between the TGN/EE and osmotic stress. This is why we believe our findings are novel and highly interesting. Here we aimed to present the connection between salt and osmotic stress, TGN/EE ion and pH regulation and CCC1. It will be exciting to build on these findings to detail the connection between osmotic stress and pH changes in the TGN/EE lumen in the coming years in future studies.

We have re-written the discussion to make our take home messages more prominent.

The authors sometimes make statements regarding that one sample /tissue is brighter than another, however I do not think these types of qualitative statements are appropriate, and should be supported by quantitative data or removed.

In our manuscript we make comments that BFA bodies are often more compact and due to this, appear brighter in wildtype compared to *ccc1*; this is visible in the images provided. We make these comments to be transparent about the samples and that the images are accurate in depicting *ccc1* plants with differently formed BFA bodies. We had not commented on the brightness of other images, however, as requested above, we have now included a quantification of PIN2-GFP at the PM.

FiguresScale bar sizes should not be shown in the image, but rather described in the figure legend, as otherwise it is too small to see.

We have amended the figures.

Figure 4.The quantification of panel C is in B. The order should be switched of these two panels. (Same as E and F).

The quantification of the collet hair phenotypes now is placed after the images.

The manuscript would flow better by starting with the clear cell elongation defects and rescue in panel G and H. The panel legend may also need to be changed to root cell elongation and cell identity defects.

We appreciate this suggestion by the reviewer and the authors have discussed the potential to switch the results as suggested by the reviewer but the majority preferred the current flow. We have concentrated on the macro-phenotypes first and then shown the basis for this by examination elongation.

Figure 5. (D) Cannot see any differences between the WT and mutant. A heatmap LUT may better show differences in image intensity.

The images are now presented with a ‘fire’ LUT.

Figure 7. (A) This image is way too small to see any differences between the two genotypes. A zoomed in image would be better.

Zoomed-in images are now provided in Figure 7A. For the other images, the colour differences are the most important feature where structures are not distinguishable, even when the images are enlarged due to resolution constraints in these highly processed images.

Reviewer #2:In the submitted paper, the authors first show that activity of the CCC1 promoter is ubiquitous. They further analyze the phenotype of the mutant in the root and show a root cell elongation defect in epidermal cells as well as in root hairs. The ccc1 mutants also lack the collet root hairs and show trichoblast-atrichoblast cell fate identity defects in the primary root. The authors perform a set of elegant experiments where they show that, surprisingly, the ccc1 plants are resistant to hyperosmotic environment. The ccc1 cells show delayed plasmolysis, ccc1 seeds show better germination, and ccc1 root hair elongation is recovered in hyper-osmotic media. Interestingly, the absence of collet root hairs was also recovered in hyper-osmotic environment, even though it is not clear whether this was caused by 'reparation' of collet hair elongation or collet hair cell fate specification. The phenotypic analysis is carefully performed and the results are unexpected and intriguing.The authors further show that in root trichoblasts, GFP-CCC1 localizes to the TGN/EE compartment, and that in this tissue, the fusion protein recovers the root hair elongation of the mutant. Further, the authors focus on the subcellular phenotypes of the endomembrane system performance in the ccc1 mutant background. It is shown that PIP2 aquaporin internalized less in the ccc1 than in control, which hints to that endocytosis is reduced in the ccc1 cells. An alternative explanation however could be that the mutant is more osmotically tolerant also on the subcellular level. To test the endomembrane trafficking rate, PIN2 aggregation and recovery from BFA bodies is performed, as well as quantifications of FM4-64 uptake, and the authors conclude that the mutant has generic endomembrane trafficking defects.The authors hypothesize that the endomembrane defects might stem from a disturbance in TGN/EE luminal pH caused by an ion imbalance in the ccc1 cells. Therefore, they measured the luminal pH of TGN/EE using a genetically encoded phluorin and demonstrate a more alkaline pH values in the ccc1 mutants. Finally, the authors show that during hyperosmotic stress, the TGN/EE pH rises in the control plants, suggesting that this pH rise is functionally connected to the stress response. The second part of the manuscript that focuses on subcellular phenotypes uses advanced live-cell imaging tool and successfully measures pH in minute volumes of TGN/EE compartments. In addition, the specificity of the phenotype is demonstrated by careful analysis of vacuolar and cytoplasmic pH. These well performed experiments indeed point to the function of CCC1 in ion control in TGN/EE.

Many thanks for your positive comments on our work. We are glad it is appreciated.

Weaknesses of the manuscript:The functionality of the GFP-CCC1 fusion is questionable as it was impossible to obtain transgenic lines that would express GFP-CCC1 under the control of the native promoter, not allowing full complementation of the ccc1 phenotype. This hints to a possible dominant-negative effect of this particular protein fusion. The authors therefore express GFP-CCC1 using a trichoblast-specific promoter and show that the root hair elongation phenotype is complemented, demonstrating some functionality of this construct. Moreover, the root hair length data in the ccc1-1 mutants shown in figure 2D and 3C differ, which to some extent weakens the important conclusion that the GFP-CCC1 is functional at least in this cell type. Functionality of this construct is a crucial aspect for the manuscript. The possible dominant-negative effect of the construct weakens the conclusion about the subcellular protein localization, which in turn weakens the main conclusion of the paper – that CCC1 by regulating ion fluxes in the TGN/EE allows proper endomembrane functionality.

The reviewer notes the observed difference in wildtype root hair length between experimental data shown in Figures 2D and 3C. Yes, this is correct. The experiments were done several months apart (different to the biological replicate experiments pooled for one graph, which were always conducted around the same time). Root hairs are highly reactive to environmental conditions and as such, plants grown at different times of the year commonly have differences in root hair length. As such, comparisons can only be made to a control, grown at the same time, when looking for differences in treatments or genotypes. Therefore, the data from 2D and 3C cannot be compared to each other quantitatively but rather, qualitatively.

In regards to the localisation and constructs used, neither N- nor C- terminally tagging produces transformants. Our experimental approaches suggest that all terminal tagging of CCC1 is dominant negative if the construct is expressed from the embryo stage (native promoter or ubiquitous promoters such as 35S). Expression of untagged CCC1 by either the native or 35S promoter rescues the phenotype of *ccc1* KO plants. We have now provided a table (Supplementary File 1a) that summarises all attempts to localise CCC1 and the outcome, including the generation of an antibody.

We have additionally added details on previous proteomics studies, which identified CCC1 as a high-confidence TGN/EE resident protein (L186 and discussion)

The subcellular localization of CCC1 should be demonstrated without any doubts, as it was previously localized to the PM and endomembranes in pollen tubes (Domingos 2019). If CCC1 localized at the PM, alternative explanations of the phenotypes of the nature of the mutant phenotype that would include regulation ion fluxes across the PM would appear more probable than the TGN/EE hypothesis.

The reviewer highlights that CCC1 has been proposed to localise to the PM in pollen tubes in Domingos et al. (2019). Localisation of CCC1 shown in Domingos et al. (2019), lacks colocalisation with a marker and importantly, no complementation of the knockout phenotype is shown with the tagged protein. We have added a paragraph in the discussion on this topic (L488-507).

Quantification of endomembrane trafficking represents another important argument in the proposed hypothesis. The section that demonstrates the reduction in exo- and endocytosis is however not utterly convincing. It has been shown that a major contribution to the BFA body PIN2 pool originates from de-novo synthesis of the protein (Jasik et al., PMID:27506239). In the figure 6A, it is apparent that the BFA washout leads to disappearance of BFA bodies in the ccc1 mutant, but the level of PM fluorescence was decreased, leading to an apparent 'minimal recovery' of the cytoplasm:PM ratio. In case of endocytosis, the experiment combining FM4-64 uptake with BFA is hard to interpret as endocytosis visualization, because TGN/EE aggregation might be disturbed in the ccc1, as the authors suggest. A more detailed endomembrane trafficking then simple cytoplasm/PM ratios of signal could be performed to address what is happening with trafficking in this interesting mutant.

In the FM4-64 experiment, there is a poor formation of BFA bodies and a lower ratio in *ccc1*, however, despite having the same poor formation of BFA bodies in the PIN2-GFP experiment, the ratio is higher, indicating that the visibility of BFA bodies is not crucial to the accumulation or measurement of fluorescence (discussed in L381). The reviewer does rightly state that *de-novo* protein synthesis is a major contributor to intracellular protein accumulation in these assays and that is why the assay focuses on the rate of recovery. That is, we measured PIN2-GFP recovery to the PM regardless of the origin of the PIN2-GFP protein in BFA bodies (de novo or from endocytosis).

Further characterisation of the impacts TGN/EE luminal pH changes have on endomembrane trafficking is undoubtedly an interesting topic of study, as highlighted by the reviewer’s comments. The work detailed in this manuscript is focused on assessing the role of CCC1 in TGN/EE pH and ion regulation. As such, the results presented will enhance the potential to investigate the impact of TGN/EE pH on endomembrane trafficking by providing details of another tool, *ccc1*, which can be used to investigate this link and by further detailing the impact of environmental conditions on TGN/EE pH. It is not the aim of this study here to investigate the link between TGN/EE pH and endomembrane trafficking and as such, we believe that further results detailing endomembrane trafficking will detract from the central results of this work. However, this aspect highlights the general interest and importance of our work for other fields of plant science, and to other fields as CCC proteins are present in all organisms.

Suggestions:It might be helpful to analyze the subcellular localization of CCC1-GFP in the vegetative tissues in the Lat52-driven constructs (Domingos 2019). Lat52 can be leaky in the vegetative tissues and has been used to drive mild expression of proteins such as T-PLATE (PMID: 24529374). This could help strengthen the claim that CCC1 is localized to TGN. The Lat52 driven construct in the Domingos paper was constructed from genomic sequence, contains introns, which is another difference from the CDS construct used in the submitted manuscript.

The reviewer suggests that the use genomic *CCC1* sequence rather than the CDS sequence may aid in the difficulties expressing tagged CCC1 in plants. While we appreciate the thoughtful suggestion, expression and complementation of *ccc1* phenotypes with an untagged, CDS-derived, CCC1 using the native promoter suggests that introns are not the cause of the lack of transformants but rather, that it is the tagging. The localisation presented in Domingos et al. (2019) lacks an investigation into functionality or complementation of the construct used and a quantitative colocalization with a marker is not performed. In addition, it is unclear how putative *LAT52* driven expression in vegetative tissue would provide further insight compared to *EXP7* driven expression in vegetative cells which complements a documented phenotype. As described in the response to reviewer 1, we have now included a table (Supplementary File 1a) summarising CCC1 constructs, and have added a paragraph in the discussion on CCC1 localisation in pollen tubes.

We have contacted the Feijo lab (corresponding author on the Domingos paper) and no seed for LAT52pro:CCC1-GFP is available. Further, they have not observed expression of protein in vegetative tissue driven by *LAT52*. And while it might be sometimes of some use, we think using a pollen specific promoter to drive putative leaky vegetative tissue expression would be a contrived and difficult to interpret experiment, which does not allow any control over the cell or tissue type. We respectfully suggest, that such an experiment is likely to attract more criticism than the approach we chose.

What is the topology of the CCC1 in the membrane – where are the GFP fusions localized? Could the problem of the non-shining internal GFP fusions be the low luminal (or apoplastic in the case of PM localization of the protein) pH of the compartment that would quench the GFP fluorescence? This could be tested by rising the pH of the endomembranes of the transgenic lines, or by using a pH-insensitive fluorescent protein.

The structure of an animal CCC protein has been experimentally determined, the zebrafish NKCC1. NKCC1 has a complex membrane topology, and forms very compact homodimers, with intertwined C-terminal domains. It is likely that CCC1 shares the complex topology found in the zebra fish NKCCC1.

We have added clarification on how the internal tagging sites were selected (L200 and Supplementary File 1a).

Comments:Student t-test for comparing multiple samples is repeatedly used. Appropriate statistical tests for comparing multiple samples should be used instead, and also the data distribution should be tested for normality in cases when parametric tests are used.

We have replaced the student t-test in all graphs with multiple sample tests e.g. ANOVAs.

The authors explain the short root phenotype by a reduction of cell length. However, from the data provided, the cell length seems to be far less reduced than the overall root growth.

We agree and have amended the text (L167).

Trichoblast identities: Is the trichoblast identity problem complemented in the trichoblast-CCC1-expressing lines?

Thank you for picking up on this. It is not, and cannot expected to be, as trichoblast identity is set prior to EXP7 expression. We have now provided data of the trichoblast marker *PRP3::H2B-2xmCherry*, expressed in wildtype and *ccc1* (Figure 2 – supplementary figure 1C).

Quantification of endocytosis and endomembrane trafficking is not very clear to me: how can the cytoplasm/PM ratio of fluorescence give the value of 1, when there is almost no fluorescence in the cytoplasm in the 10 min of FM4-64 uptake or in the T0 in case of the PIN2 GFP fluorescence?

The reviewer is correct that a ratio of 1 would be highly unusual. The value of 1 is not absolute but rather, relative to wildtype under untreated conditions. The legend text is now revised to clarify this further.

Also, not always same areas of roots are shown for treatments and mutants (authors state that data was quantified in elongation zone, but images show rather meristem/transition zone). Were the quantifications of the fluorescence signals done blindly? Do not take this as an offense, but from my experience, in case of data, where regions of interest are drawn by hand, experiments should be done so that the researcher imaging the samples at the microscope does not know which genotype they are imaging, as the FM signal in different root zones and z sections is very variable.

Measurements were taken in the late-transition/early elongation zone of roots and from a broader range than displayed in the figures where images are cropped and zoomed to enable better clarity. Cells can have a slightly different morphology in *ccc1* plants where they tend to be wider than in wildtype.

Measurements were not done blind. This is because it is not possible to do so due to the obvious defects in *ccc1* plants which make them very easy to identify (on the plant level, and on the image level). We are aware of the challenge of unconscious bias when measuring, which is why we decided to measure a large number of cells from each root from a broad range of zones from late-transition through to the elongation zone in both wildtype and *ccc1* to reduce the chance of unconscious bias towards a specific cell type and stage in this area of the root.

I would suggest a simple pulse-chase FM4-64 uptake experiment without including BFA to reveal the dynamics of FM uptake in the mutant to provide more convincing data showing the endocytotic defect.

We have added this experiment (Figure 6C and D).

[Editors’ note: what follows is the authors’ response to the second round of review.]

We remain enthusiastic about your work and welcome you to submit a revised manuscript along the lines of what you outlined in your rebuttal. Our original decision was based on the expected time needed to perform more conclusive experiments, but you have convinced us that many of our concerns can be addressed with thorough discussion, more precise language, and a few experiments that will only take 1-2 months.Regarding our main concerns:Functionality: As you could not obtain transformants in most cases, we still think this unfortunately means that tagged CCC1 is not fully functional. However, we also agree that your focus on the complementation of a cellular phenotype is sound and justified. We would ask that you do provide a more thorough cataloging of complemented and non-complemented phenotypes (you mentioned in your letter that these data are available, at least with respect to trichoblast identity), and further discussion on this construct in particular, and in comparison to others, as outlined in your rebuttal. In addition, a more detailed discussion of the issue and a more detailed overview (in a Supplemental table, as you suggest) of what has been tried to obtain the functional transformants would demonstrate that everything possible was tried. In addition, a more pointed discussion about the localization of CCC1 as found in proteomics studies should be included.

As suggested, to highlight and explain why we used the construct we did, we have now added a supplementary table (Supplementary File 1a) summarising all approaches tried to obtain a functional CCC1 construct for localisation, and describe these approaches in the Results section (L189-201). Further, we now also mention the results of previous proteomic studies, that identified CCC1 as a high-confidence TGN/EE resident protein in the introduction and results (L86-90 and L182-186). Additionally, we added a paragraph highlighting the advantages of the construct we use to show functional complementation of localised CCC1 in comparison to other approaches previously used for CCC1 localisation; and discuss the limitations of all the approaches, including the construct used in our study (L491-504). The new data combined with the originally submitted data consistent with the EXP7 promoter driving expression of functional CCC1 in trichoblast cells after cellular identity was established.

As requested, we have now included additional phenotyping of *ccc1* plants expressing GFP-CCC1. Phenotypes now documented include trichoblast patterning (not complemented, Figure 3 – supplementary figure 1B), trichoblast length (complemented, Figure 3 – supplementary figure 1C) and formation of collet hairs (not complemented, Figure 2 —figure supplement 1A), in addition to the existing root hair length results (complemented, Figure 3B and C). We now also include new data obtained using a trichoblast marker in WT and *ccc1*. This shows that the ectopic root hairs in *ccc1* express this marker (Figure 2 —figure supplement 1C) suggesting that cell identity is similar to root hair cells in the correct cell files. The results on collet hairs highlights that these hairs have a different identity and are not complemented through use of the EXP7 promoter.

We thank you for this guidance. The inclusion of all this requested information provides further evidence supporting our conclusions and has improved our manuscript.

Endocytic trafficking: We agree that a more careful wording will help clarify distractions and focus on the function of CCC1 in the TGN/EE and its role in ion regulation. However, the data presented to determine defects in endocytic trafficking may still need some re-analysis: for example, quantification of PM and BFA signals. To help clarify the defect in trafficking, it would be useful to do a FM pulse chase experiment to complement the approach you show in Figure 6E-F. One reviewer suggests that the simplest way to improve this section would be to remove Figure 6C 10 min time point (BFA + FM) and replace this with an FM pulse-chase experiment (FM treatment for 10-15 min without BFA) to test for defects in endocytosis/trafficking more clearly. Quantification of the plasma membrane signal in Figure 6A alone would also help with the interpretations of the BFA washout experiment (i.e., if the PIN2-GFP signal is significantly reduced in the mutant this may be an interesting result as well).

Thank you for this constructive feedback. We have added the pulse chase experiment in Figure 6. We found no difference between the WT and *ccc1* after a 10 min treatment with FM4-64 without BFA, which supports our hypothesis that CCC1 affects endocytic trafficking but not endocytosis. We have kept the 10 min FM4-64 with BFA treatment in the Figure, as we thought it serves as a good control for the 60 min time point of the same experiment. The data now nicely demonstrates that there is likely no difference in endocytosis (10 min without BFA) but there is a difference in endocytic trafficking (60 min + BFA).

Furthermore, as suggested, we have included quantification of PM PIN2-GFP signal and found that there is a reduction of the signal in the mutant (Figure 6 – supplementary figure 1A). This further supports the hypothesis that CCC1 is part of the ion and pH regulating network in the TGN/EE, since similar results have been found for the other proteins involved. Importantly, we have also included an additional graph (Figure 6 – supplementary figure 1B) to highlight that the PM:cytosol signal ratios are not impacted by the lower abundance of PIN2 in the mutant, which validates our approach to investigate endocytic trafficking.

In addition, we noticed that the request on data summarising the outcome of *ccc1* crosses with other mutants affected in TGN/EE pH was not part of the public reviewer comments, which we address below, but was part of the previous editorial reply. We have added a supplementary table (Supplementary File 1b) detailing the outcome of the crosses, including information on which mutant lines were used, and a new supplementary Figure (Figure 7 – supplementary figure 3) showing the one, infertile triple mutant obtained.